# MINIMUM-EXCESS-WORK GUIDANCE

## ABSTRACT

We propose a regularization framework inspired by thermodynamic work for guiding pre-trained probability flow generative models (e.g., continuous normalizing flows or diffusion models) by minimizing excess work, a concept rooted in statistical mechanics and with strong conceptual connections to optimal transport. Our approach enables efficient guidance in sparse-data regimes common to scientific applications, where only limited target samples or partial density constraints are available. We introduce two strategies: Path Guidance, which facilitates sampling of rare transition states by concentrating probability mass on user-defined subsets, and Observable Guidance, which aligns generated distributions with experimental observables while preserving entropy. We demonstrate the framework's versatility on two coarse-grained protein models, highlighting its ability to sample transition configurations and to correct systematic biases using experimental data. The method bridges thermodynamic principles with modern generative architectures, offering a principled, efficient, and physics-inspired alternative to standard fine-tuning in data-scarce domains. Empirical results highlight improved sample efficiency and bias reduction, underscoring its applicability to molecular simulations and beyond.

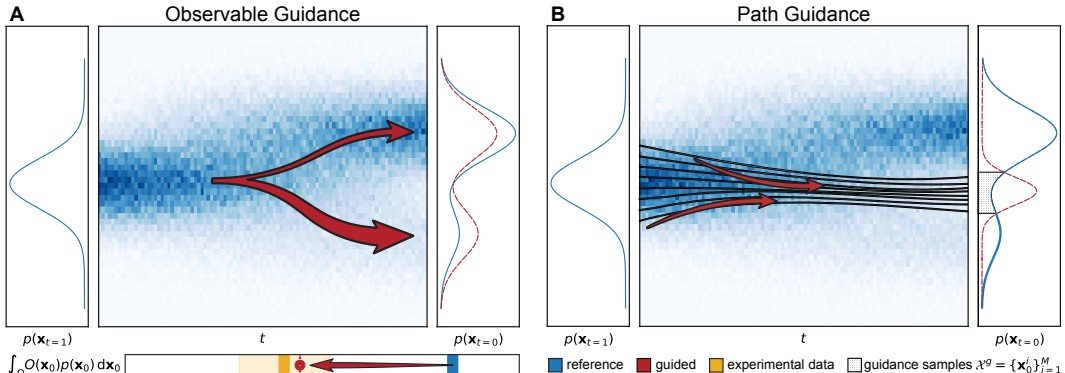

Figure 1: Schematic comparison of observable and path guidance. Both panels show the evolution of probability density over time $t$ (blue heat map) with marginal distributions $p(\mathbf{x}_{t=1})$ and $p(\mathbf{x}_{t=0})$ on the sides (blue: reference model, red dashed: guided model). (A) Observable guidance perturbs the score function (red arrows) to match experimental observables (yellow) with unknown data distribution $p(\mathbf{x}_{t=0})$ using minimal excess work. (B) Path guidance steers sampling trajectories (black solid) toward specific regions defined by guidance samples $\mathcal{X}^g = \{\mathbf{x}_0^i\}_{i=1}^M$ (dotted grey).

## 1 INTRODUCTION

Probability flow generative models, such as normalizing flows (Papamakarios et al., 2021; Liu, 2022; Albergo et al., 2023; Lipman et al., 2023) and diffusion models (Song et al., 2020; Ho et al., 2020), enable the modeling of complex, high-dimensional data distributions across a wide range of applications. These models generate samples through the integration of differential equations evolving a tractable distribution into an approximation of the data distribution. Although these models excel at general distribution learning, many scientific applications require precise control over generated samples to meet sparse observational constraints (e.g., limited transition state configurations or partial

density constraints from experiments). Current guidance methods struggle in data-scarce regimes as they typically rely on either specialized training or abundant reward signals. Existing approaches often involve fine-tuning (Wallace et al., 2024; Black et al., 2023; Domingo-Enrich et al., 2024), incorporating conditional information during training (Ho & Salimans, 2022; Nichol et al., 2021), or training an additional noise-aware discriminative model (Song et al., 2020).

While effective, these methods may be impractical in sparse-data settings as the number of tuned parameters can be large, even when using parameter-efficient adapters. This motivates a new approach in which we apply minimal perturbations to the trained model, enabling controlled generation under very sparse constraints. Inspired by statistical mechanics, we introduce an approach for regularizing guidance of probability flow generative models based on the principle of minimum excess work (MEW). In this context, "work" is a measure of the physical effort, e.g., energy, needed to transform a system from one macrostate to another, where a macrostate is characterized by a probability density function. MEW thereby acts as a natural, physics-inspired regularization scheme for guiding generative models. We develop the theoretical framework for MEW-based regularization of generative models, explicitly connecting it to optimal transport theory, and validate its effectiveness through extensive benchmarks across multiple scales and systems. In addition to introducing the MEW framework, we propose a simple yet effective form of path guidance tailored to sparse sampling problems. We specialize MEW guidance to two common challenges in molecular simulation. First, **Observable Guidance**: a bias-correction method that matches experimental observables while preserving the entropy of the reference ensemble via a minimum-excess-work regularizer. We validate this approach on a toy system and two coarse-grained protein Boltzmann emulators. With this approach, we thus correct the systematic bias in the base model and are therefore able to improve the prediction of unmeasured observables since they report on the same thermodynamics. Second, **Transition-State Sampling**: a path-guidance-based sampling strategy that concentrates samples on user-specified regions, e.g., the low-probability transition region between states, which we evaluate on the coarse-grained Boltzmann emulator.[1]

## 2 BACKGROUND AND PRELIMINARIES

**Diffusion models** learn a stochastic process that maps a simple prior distribution $p_1$ to an approximation $p_0$ of the data distribution $q_0$. This is typically done by reversing a known noising process governed by an Ornstein–Uhlenbeck SDE, $\mathrm{d}\mathbf{x}_t = \mathbf{f}(\mathbf{x}_t, t)\,\mathrm{d}t + g(t)\,\mathrm{d}\mathbf{w}_t$ with $\mathbf{f}(\mathbf{x}_t, t)$ linear in $\mathbf{x}_t$. This process induces a family of marginals $q_t$ with simple forward transitions $q_t(\mathbf{x}_t|\mathbf{x}_0) = \mathcal{N}(\mathbf{x}_t; \alpha_t \mathbf{x}_0, \sigma_t^2 \mathbf{I})$, where $\alpha_t, \sigma_t$ are determined by the SDE coefficients. Given $q_1$ and the score $\nabla_{\mathbf{x}_t} \log q_t(\mathbf{x}_t)$, one can sample from $q_0$ via the time-reversal (Anderson, 1982):

$$\mathrm{d}\mathbf{x}_t = [\mathbf{f}(\mathbf{x}_t, t) - g(t)^2 \nabla_{\mathbf{x}_t} \log q_t(\mathbf{x}_t)]\,\mathrm{d}t + g(t)\,\mathrm{d}\widetilde{\mathbf{w}}_t\,, \qquad \mathbf{x}_1 \sim q_1\,, \tag{1}$$

where $\widetilde{\mathbf{w}}_t$ is a reverse-time Wiener process, or via the *probability flow ODE* (Maoutsa et al., 2020; Song et al., 2020):

$$\frac{\mathrm{d}\mathbf{x}_t}{\mathrm{d}t} = \mathbf{f}(\mathbf{x}_t, t) - \frac{1}{2} g(t)^2 \nabla_{\mathbf{x}_t} \log q_t(\mathbf{x}_t)\,, \qquad \mathbf{x}_1 \sim q_1\,, \tag{2}$$

both having the same time-marginals $q_t$ as the forward process. In practice, the score is approximated by a score model $\mathbf{s}_\theta(\mathbf{x}_t, t)$, and a simple distribution $p_1 \approx q_1$ is used as initial distribution at $t = 1$:

$$\mathrm{d}\mathbf{x}_t = [\mathbf{f}(\mathbf{x}_t, t) - g(t)^2 \mathbf{s}_\theta(\mathbf{x}_t, t)]\,\mathrm{d}t + g(t)\,\mathrm{d}\widetilde{\mathbf{w}}_t \qquad \mathbf{x}_1 \sim p_1 \tag{3}$$

$$\frac{\mathrm{d}\mathbf{x}_t}{\mathrm{d}t} = \mathbf{f}(\mathbf{x}_t, t) - \frac{1}{2} g(t)^2 \mathbf{s}_\theta(\mathbf{x}_t, t) \qquad \mathbf{x}_1 \sim p_1 \tag{4}$$

We will denote by $\{p_t\}_{t \in [0,1]}$ the probability path induced by Eq. (3) or Eq. (4).

**Equilibrium sampling of the Boltzmann distribution.** A key challenge in statistical mechanics is to generate independent samples from the Boltzmann distribution

$$\mathbf{x} \sim p(\mathbf{x}) \propto \exp(-\beta U(\mathbf{x})), \tag{5}$$

where $\beta = (k_\mathrm{B} T)^{-1}$ is the inverse temperature and $U(\mathbf{x})$ is the potential energy of a configuration $\mathbf{x} \in \Omega \subseteq \mathbb{R}^d$. This distribution underlies estimation of macroscopic observables, such as $\mathbb{E}_{p(\mathbf{x})}[O_i(\mathbf{x})]$,

---

[1]Code is available at https://anonymous.4open.science/r/MEW-Guidance-F761

which allow for a direct comparison to experimental data. However, sampling from $p(\mathbf{x})$ is notoriously difficult due to the rugged energy landscape $U(\mathbf{x})$. Traditional methods such as Molecular Dynamics (MD) or Markov Chain Monte Carlo (MCMC) suffer from slow mixing and generate highly correlated samples that often fail to cross energy barriers between metastable states. This leads to biased estimates and poor coverage of transition configurations, i.e., regions in state space that are severely undersampled but mechanistically crucial. Recent work on Boltzmann Generators (Noé et al., 2019; Klein et al., 2024b; Midgley et al., 2023; Köhler et al., 2020; Moqvist et al., 2025; Tan et al., 2025) addresses these challenges by learning direct mappings from simple priors to Boltzmann-like distributions. Yet, two issues remain: inaccuracies in the potential energy model can bias the learned distribution (Kolloff et al., 2022; Klein et al., 2024a), and physically important but low-probability states (e.g., transition states) remain exponentially rare. In this work, we address both problems by guiding a generative model using sparse experimental or structural information, leveraging a coarse-grained Boltzmann emulator inspired by Arts et al. (2023) and show how our method can be integrated into a state-of-the-art Boltzmann emulator (Lewis et al., 2024) to sample protein ensembles consistent with experimental data.

**Maximum Entropy Reweighting** is a broadly adopted technique to overcome force field inaccuracies in potential energy models (Pitera & Chodera, 2012; Cavalli et al., 2013; Olsson et al., 2013; 2016; Boomsma et al., 2014; White & Voth, 2014; Beauchamp et al., 2014; Hummer & Köfinger, 2015; Bonomi et al., 2016). The result of this optimization is a tilted distribution which depends on a set of Lagrange multipliers, $\{\lambda_i\}$, each corresponding to an experimental observable of interest. The solution $p'(\mathbf{x}) \propto p(\mathbf{x}) \exp\left(- \sum_{i=1}^{M} \lambda_i O_i(\mathbf{x})\right)$ minimizes the KL divergence from the reference distribution $p(\mathbf{x})$, subject to the constraints $\mathbb{E}_{p'(\mathbf{x})}[O_i(\mathbf{x})] = o_i$. A detailed derivation is provided in Appendix A.1 for the reader's convenience. However, until now, this approach has been limited to reweighting fixed sets of samples $\mathcal{X} = \{\mathbf{x}_i\}_{i=1}^{M}$ (e.g., an MD trajectory), thus motivating methods to apply these principles in a generative setting.

**Loss Guidance** is the process of adjusting the diffusion process to satisfy a target condition $\mathbf{y}$ without fine-tuning and has been explored in several prior works (Bansal et al., 2023; Chung et al., 2023; Song et al., 2023). To sample from the conditional distribution $p(\mathbf{x}_0|\mathbf{y})$ *post hoc*, we can use the following identity: $\nabla_{\mathbf{x}_t} \log p(\mathbf{x}_t|\mathbf{y}) = \nabla_{\mathbf{x}_t} \log p(\mathbf{x}_t) + \nabla_{\mathbf{x}_t} \log p(\mathbf{y}|\mathbf{x}_t)$. Obtaining $\nabla_{\mathbf{x}_t} \log p(\mathbf{y}|\mathbf{x}_t)$ typically requires training a separate model on the noisy states $\mathbf{x}_t$, as done in classifier guidance (Song et al., 2020). Alternatively, the posterior mean $\hat{\mathbf{x}}_t(\mathbf{x}_t) := \mathbb{E}_{p(\mathbf{x}_0|\mathbf{x}_t)}[\mathbf{x}_0]$ can be used as an estimate of the clean data $\mathbf{x}_0$. Using Tweedie's formula, the posterior mean can be expressed as $\mathbb{E}_{p(\mathbf{x}_0|\mathbf{x}_t)}[\mathbf{x}_0] = \frac{1}{\alpha_t}(\mathbf{x}_t + \sigma_t^2 \nabla_{\mathbf{x}_t} \log p(\mathbf{x}_t))$. This allows us to approximate the likelihood in data space via $\log p(\mathbf{y}|\hat{\mathbf{x}}_t(\mathbf{x}_t)) \simeq \ell(\mathbf{y}, \hat{\mathbf{x}}_t(\mathbf{x}_t))$, where $\ell$ denotes a suitable differentiable loss function (e.g., cross-entropy or log-likelihood under a differentiable model). The gradient $\nabla_{\mathbf{x}_t} \log p(\mathbf{y}|\hat{\mathbf{x}}_t(\mathbf{x}_t))$ can then be computed by backpropagation. In practice, the mean is approximated using the score model $\mathbf{s}_\theta(\mathbf{x}_t, t)$, allowing the score estimate to be updated as $\nabla_{\mathbf{x}_t} \log p(\mathbf{x}_t|\mathbf{y}) \simeq \mathbf{s}_\theta(\mathbf{x}_t, t) + \eta_t \nabla_{\mathbf{x}_t} \ell(\hat{\mathbf{x}}_t(\mathbf{x}_t), \mathbf{y})$ with $\eta_t$ being a guiding strength function.

**Work and Optimal Transport.** In statistical mechanics, thermodynamic work $W$ is the energy required to transform a system from a probabilistic state $p$ to another $p'$. For a continuum system:

$$W = \iint \mathbf{J}(\mathbf{x}, t) \cdot \mathbf{F}(\mathbf{x}, t)\, \mathrm{d}\mathbf{x}\, \mathrm{d}t, \qquad (6)$$

where $\mathbf{J}(\mathbf{x}, t) = \mathbf{v}(\mathbf{x}, t)p_t(\mathbf{x})$ is the probability flux and $\mathbf{F}(\mathbf{x}, t)$ is the force applied to the system. This generalizes the classical work expression $W = \int F(\mathbf{x})\, \mathrm{d}\mathbf{x}$. When the force and velocity field coincide (i.e., the Jacobian of the push-forward map associated with the velocity field is a diffeomorphism), they can be expressed as spatial gradients of a potential $u(\mathbf{x}, t)$ (Brenier, 1991). Under these conditions, $W$ becomes equivalent to the kinetic energy in the Benamou–Brenier formulation of optimal transport (Benamou & Brenier, 2000), and provides an upper bound on the squared 2-Wasserstein distance between the distributions:

$$W_2^2(p, p') \leq \iint \|\mathbf{v}(\mathbf{x}, t)\|^2 p_t(\mathbf{x})\, \mathrm{d}\mathbf{x}\, \mathrm{d}t = W \qquad (7)$$

where $\mathbf{v}$ and $p$ satisfy $\frac{\partial}{\partial t} p_t(\mathbf{x}) = -\nabla_{\mathbf{x}} \cdot [p_t(\mathbf{x})\, \mathbf{v}(\mathbf{x}, t)]$. Minimizing $W$ yields the optimal transport map that transforms $p$ into $p'$ along the path requiring minimal energy. The idea of identifying probability paths minimizing the kinetic energy, or more generally a Lagrangian, has recently been applied to improve the efficiency of probability flow generative models (Tong et al., 2020; 2023; Klein et al., 2023; Irwin et al., 2025; Shaul et al., 2023; Albergo et al., 2023; Neklyudov et al., 2023a;b).

## 3 MINIMUM-EXCESS-WORK GUIDANCE

During the generative process, we transform a simple distribution $p_1 \sim \mathcal{N}(\mathbf{0}, \mathbf{I})$ into a complex data distribution $p_0$ with support $\Omega \subseteq \mathbb{R}^d$ by solving the reverse-time SDE (3) or the ODE (4). To incorporate additional constraints and align the generative process with new information, we modify the drift of Eqs. (3) and (4) by introducing an additive perturbation to the score model:

$$d\mathbf{x}_t = \left( \mathbf{f}(\mathbf{x}_t, t) - g(t)^2 \left[ \mathbf{s}_\theta(\mathbf{x}_t, t) + \mathbf{h}_\vartheta(\mathbf{x}_t, t) \right] \right) dt + g(t) d\widetilde{\mathbf{w}}_t \qquad \mathbf{x}_1 \sim p_1 , \tag{8}$$

$$\frac{d\mathbf{x}_t}{dt} = \mathbf{f}(\mathbf{x}_t, t) - \frac{1}{2} g(t)^2 \left[ \mathbf{s}_\theta(\mathbf{x}_t, t) + \mathbf{h}_\vartheta(\mathbf{x}_t, t) \right] \qquad \mathbf{x}_1 \sim p_1 , \tag{9}$$

where $\mathbf{h}_\vartheta : \mathbb{R}^d \times [0, 1] \to \mathbb{R}^d$ is a time-dependent vector field.

**The aim of MEW guidance** is to satisfy a guidance objective for the guided distribution $p_0' \neq p_0$, while minimizing the *excess work* associated with $\mathbf{h}_\vartheta(\mathbf{x}_t, t)$, required to modify the probability density, $p_0$. We define the excess work in the context of an unperturbed and perturbed system described by the following ODEs over $t \in [0, 1]$ with $p_1 = p_1'$:

$$\frac{d\mathbf{x}_t}{dt} = \mathbf{v}(\mathbf{x}_t, t) , \qquad \frac{d\mathbf{x}_t}{dt} = \mathbf{v}(\mathbf{x}_t, t) + \mathbf{u}(\mathbf{x}_t, t) , \tag{10}$$

with the respective time-marginal densities $p_t, p_t'$. Loosely following Eq. (7), we define the excess work as $\Delta W := \iint \|\mathbf{u}(\mathbf{x}, t)\|^2 p_t'(\mathbf{x}) \, d\mathbf{x} \, dt$, which for the ODEs (4) and (9) becomes:

$$\Delta W(\vartheta) = \iint \frac{g(t)^4}{4} \|\mathbf{h}_\vartheta(\mathbf{x}, t)\|^2 p_t'(\mathbf{x}) \, d\mathbf{x} \, dt . \tag{11}$$

To justify our choice of excess work as a regularizer, it is helpful to understand how perturbations affect the generated distribution. In particular, we would like $p_0'$ to remain close to the reference base distribution $p_0$. While stability bounds of this type have appeared in the literature on ODEs and SDEs, we restate tailored versions here for completeness, with proofs in Appendices A.2 and A.3.

**Proposition 3.1.** *Let $p_t$ and $p_t'$ be the distributions at time $t$ obtained by solving the ODEs (4) and (9) backwards in time from the same initial distribution $p_1$ at $t = 1$. Assume that the vector fields are measurable in time and $L_t$-Lipschitz in space with $L_t$ integrable. Then:*

$$W_2^2(p_0, p_0') \le \int_0^1 w_{\mathrm{W}}(t) \frac{g(t)^4}{4} \mathbb{E}_{\mathbf{x} \sim p_t'} \left[ \|\mathbf{h}_\vartheta(\mathbf{x}, t)\|^2 \right] dt , \qquad w_{\mathrm{W}}(t) := e^{t + 2 \int_0^t L_s \, ds} . \tag{12}$$

**Proposition 3.2.** *Let $p_t$ and $p_t'$ be the distributions at time $t$ induced by the reverse-time SDEs (3) and (8) starting from the same distribution $p_1$ at $t = 1$. Assume that both SDEs admit strong solutions, and that $\mathbb{P}' \ll \mathbb{P}$, where $\mathbb{P}, \mathbb{P}'$ are the path measures induced by the SDEs on $C([0, 1], \mathbb{R}^d)$. Then:*

$$D_{\mathrm{KL}}(p_0' \| p_0) \le \int_0^1 w_{\mathrm{KL}}(t) \frac{g(t)^4}{4} \mathbb{E}_{\mathbf{x} \sim p_t'} \left[ \|\mathbf{h}_\vartheta(\mathbf{x}, t)\|^2 \right] dt , \qquad w_{\mathrm{KL}}(t) := \frac{2}{g(t)^2} . \tag{13}$$

Since both bounds—for the KL divergence and the Wasserstein distance—are time-reweighted versions of the excess work $\Delta W$ (11), it serves as a natural choice of regularizer for guidance objectives. We then optimize the parameters $\vartheta$ of the perturbation $\mathbf{h}_\vartheta$ by minimizing the following:

$$\mathcal{L}(\vartheta) = \mathcal{L}_1(\vartheta) + \gamma \, \Delta W(\vartheta) , \tag{14}$$

where $\mathcal{L}_1(\vartheta)$ is a guidance objective, and $\gamma$ controls the regularization strength.

We now explore how this minimum-excess-work principle is applied in the two settings: (1) guidance based on expectations of observables; (2) targeted guidance towards a user-defined subspace.

**Observable Guidance.** In this section, we guide a diffusion model to align with data that reflects an expectation, using the MEW approach. Using a set of Lagrange multipliers $\Lambda = \{\lambda_1, ..., \lambda_M\}$ pre-estimated using, e.g., the algorithm outlined in Bottaro et al. (2020), we dynamically adjust the score by estimating an augmentation factor $\mathbf{h}_\vartheta$ that ensures $\left| \mathbb{E}_{p'(\mathbf{x})}[O_i(\mathbf{x})] - o_i \right| \le \epsilon$. Note that traditional reweighting techniques typically apply bias *post-hoc*, whereas our method adapts the *generative* process. We express the guidance factor as,

$$\mathbf{h}_\vartheta(\mathbf{x}_t, t) = -\eta_t(\vartheta) \sum_{i=1}^M \lambda_i \nabla_{\mathbf{x}_t} O_i(\hat{\mathbf{x}}_t(\mathbf{x}_t)) . \tag{15}$$

In the same way that a score model $\mathbf{s}_\theta(\mathbf{x}_t, t)$ approximates the gradient of the log probability, $\mathbf{h}_\vartheta(\mathbf{x}_t, t)$ is the gradient of the observable function with respect to the latent variable $\mathbf{x}_t$. The coefficients $\lambda_i$ steers the flow towards (or away from) directions favored by the experimental observable expectation, thus "adjusting" the score of the original model. The expression in Eq. (15) thus reflects the maximum entropy principle applied in a generative setting. Its amplitude is modulated by $\eta_t(\vartheta) = \eta_{\text{init}} \exp(-\kappa(1-t))$, and our optimization strategy consists of learning the parameters $\vartheta = \{\eta_{\text{init}}, \kappa\}$ of this scalar function. Note that we use the mean posterior estimation $\hat{\mathbf{x}}_t(\mathbf{x}_t)$ discussed in Section 2, instead of using $\mathbf{x}_t$ directly (Bansal et al., 2023; Chung et al., 2023). Our optimization objective is two-fold: We want to reduce the discrepancies between the model predictions and experimental data while minimizing the excess work exerted by the augmentation. The former is a supervised loss defined as

$$\mathcal{L}_1(\vartheta) = \frac{1}{M} \sum_{i=1}^{M} \left( o_i^{\text{exp}} - \mathbb{E}_{\mathbf{x} \sim p_0'}[O_i(\mathbf{x})] \right)^2, \tag{16}$$

where $o_i^{\text{exp}}$ are the experimental values and $\mathbb{E}_{\mathbf{x} \sim p_0'}[O_i(\mathbf{x})]$ denotes the expected values under the adjusted distribution $p_0'$. To balance accuracy with the principle of maximum entropy, we introduce a regularization term based on minimizing the excess work $\Delta W$. Substituting the specific form of $\mathbf{h}_\vartheta(\mathbf{x}_t, t)$ from Eq. (15) into Eq. (11), we obtain:

$$\Delta W(\vartheta) = \int_0^1 \frac{g(t)^4}{4} |\eta_t(\vartheta)|^2 \, \mathbb{E}_{\mathbf{x} \sim p_t'} \left[ \left\| \sum_{i=1}^{M} \lambda_i \nabla_\mathbf{x} O_i(\hat{\mathbf{x}}_t(\mathbf{x})) \right\|^2 \right] \mathrm{d}t. \tag{17}$$

**Path Guidance.** In this setting, we assume access to a set of guiding samples $\mathcal{X}^g = \{\mathbf{x}^i\}_{i=1}^M$, each belonging to a target subset $A \subset \Omega$ of the sampling space. Assuming that $A$ forms a coherent region rather than being scattered across distinct modes, we modify the score of the diffusion model to sample from a perturbed distribution $\mathbf{x} \sim p_0'$ that increases the likelihood of $\mathbf{x} \in A$. Since $\mathcal{L}_1$ need not be differentiable, the objective can be formulated generally as:

$$\mathcal{L}_1(\vartheta, \varphi) = 1 - \frac{1}{N} \mathbb{E}_{\mathbf{x} \sim p_0'} \left[ \mathbb{1}_{\{\mathbf{x} \in A\}} \right]. \tag{18}$$

Guiding the diffusion process towards the subset $A$ can be done by taking advantage of the probability flow ODE (4), which holds the desirable property of providing unique *latent representations* of each data point, for any time step $t$. Starting from the guiding samples, we compute their trajectories by integrating Eq. (4) forward in time, obtaining the latent representations $\mathcal{X}_t^g = \{\mathbf{x}_t^i\}_{i=1}^M$ for time $t$. The set $\{\mathcal{X}_t^g\}_{t=0}^1$ defines a trajectory of latent representations that the model must follow to ensure its samples satisfy $\mathbf{x}' \in A$. Based on this trajectory, we can define the augmentation factor as:

$$\mathbf{h}_{\vartheta,\varphi}(\mathbf{x}_t, t) \coloneqq \eta_t(\vartheta) \nabla_{\mathbf{x}_t} \log \mathcal{K}_{h_t(\varphi)}(\mathbf{x}_t, \mathcal{X}_t^g) \tag{19}$$

with $\mathcal{K}_{h_t(\varphi)}(\mathbf{x}_t, \mathcal{X}_t^g) \coloneqq \sum_{\mathbf{x}_t^i \in \mathcal{X}_t^g} \mathrm{K}_{h_t(\varphi)}(\mathbf{x}_t, \mathbf{x}_t^i)$ where K can be any differentiable kernel with time-dependent bandwidth $h_t(\varphi)$. By updating the score function using Eq. (19), we align the sampling trajectory with that of the guiding points, while regularizing the guidance strength via the same excess work penalty as in Eq. (17), now evaluated using the time-dependent KDE score $\mathbb{E}_{\mathbf{x} \sim p_t'} \left[ \|\nabla_\mathbf{x} \log \mathcal{K}_{h_t(\varphi)}(\mathbf{x}, \mathcal{X}_t^g)\|^2 \right]$. In practice, both $\eta_t(\vartheta)$ and $h_t(\varphi)$ are implemented as sigmoid functions with learnable parameters $\vartheta = (\vartheta_{init}, \vartheta_g, \vartheta_s)$ and $\varphi = (\varphi_{init}, \varphi_g, \varphi_s)$ (see Appendix B.4) and optimized for Eq. (14) using Bayesian optimization with Gaussian Processes. The use of sigmoids allows the guidance to be stronger early in the trajectory, when $\mathbf{x}_t$ is close to the Gaussian prior and the kernel signal is more stable, and weaker near $t = 0$, where the data distribution is more complex and direct guidance is less reliable.

## 4 EXPERIMENTS

We now demonstrate the application of minimum-excess-work guidance across several experimental setups. We first evaluate path and observable guidance on two toy setups and will then proceed to showcase our approach on a coarse-grained protein Boltzmann Emulator.

### 4.1 OBSERVABLE GUIDANCE

#### 4.1.1 SYNTHETIC DATA

We chose a fully controlled synthetic system to test MEW guidance. We set up a biased 1D quadruple-well diffusion model with an accessible ground-truth Boltzmann distribution (Prinz et al., 2011) using

only the expectation of a known observable (a four-component GMM) and injecting the corresponding Lagrange multiplier via Eq. (8) following Bottaro et al. (2020). This simple test system displays two key properties in molecular dynamics: multi-modality and metastability, while keeping the corresponding Boltzmann distribution is numerically accessible, allowing us to directly gauge our methods' ability to recover the unbiased distribution and to unambiguously test whether guidance alone corrects distributional bias: we observe a ten-fold KL reduction (from $0.13$ to $0.019\pm0.002$) while matching the observable and find MEW regularization is critical to prevent mode collapse and preserve distributional fidelity. See Appendix Fig. 6 and Tab. 3 for overlays and metrics; ablation experiments are reported in Appendix Fig. 14 and Tab. 4.

### 4.1.2 COARSE-GRAINED PROTEIN BOLTZMANN EMULATOR (CGBE): CHIGNOLIN

To evaluate our method on a real-world task, we apply observable guidance to guide a pre-trained cgBE to sample conformations of chignolin, a ten-residue mini-protein that serves as a standard benchmark in protein folding studies (Honda et al., 2004; Satoh et al., 2006; Lindorff-Larsen et al., 2011). Our task is to correct systematic biases in the equilibrium sampling using only experimental measurements while preserving physical validity. This is a challenging task given the high-dimensional structured space and unknown ground truth distribution.

**Experimental Setup.** We use folding free energy $\Delta G = -k_B T \log(\frac{p_{\text{folded}}}{p_{\text{unfolded}}})$ as our observable, which captures the relative stability of different protein conformations. The reference model $p_{\text{MD}}$ shows significant bias in this metric ($-1.27$ kcal/mol vs. experimental value of $-1.87$ kcal/mol (Honda et al., 2004)), making it a suitable test case. Model architecture and training details are provided in Appendix B.3.

**Evaluation.** Our guided model achieves substantial improvements across several metrics (see Table 1) while maintaining physical validity, which we verified through the analysis of bond lengths and torsion angles (Figs. 7 to 9, in the Appendix). The guided model's folding free energy ($-1.82 \pm 0.01$ kcal/mol) closely matches the target experimental value ($-1.87$ kcal/mol), reducing mean squared error by an order of magnitude from 0.6 to 0.05 kcal/mol. Additionally,

Table 1: Quantitative metrics evaluating the guidance process: expected observables and KL divergence.

| Model $\mathcal{M}$ | $\mathbb{E}_{p_{\mathcal{M}}(\mathbf{x})}[O(\mathbf{x})]$ (kcal/mol) | $\text{KL}(p'_{\text{MD}}\|p_{\mathcal{M}})$ |
|---|---|---|
| Experimental | $-1.87$ | — |
| Reference | $-1.27$ | $0.329$ |
| Guided | $-1.82 \pm 0.01$ | $0.005 \pm 0.002$ |

the KL divergence from the reference MD trajectory improves from $0.329$ to $0.005 \pm 0.002$, demonstrating better conservation of the properties of the reference distribution, including multi-modality and entropy. Fig. 2 visualizes these improvements. Panels A and B demonstrate that guidance successfully increases the population of folded states ($d(\text{C}_1^\alpha, \text{C}_9^\alpha) < 7.5\,\text{Å}$), consistent with experimental observations. Panel C shows 50 superimposed generated structures, highlighting both the diversity and physical validity of our samples.

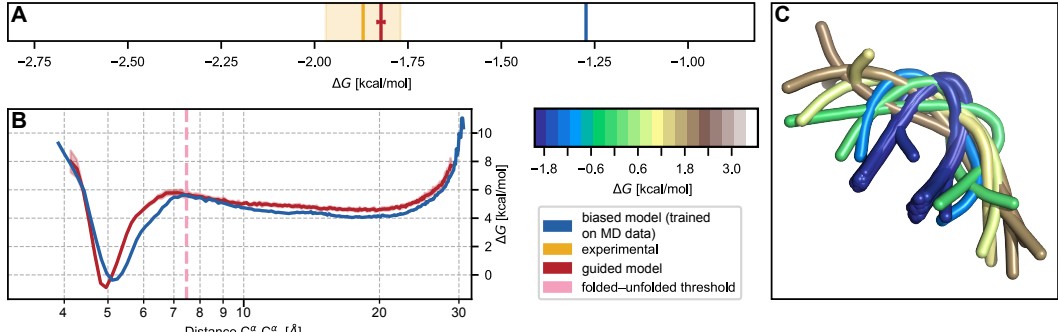

Figure 2: **Observable Guidance of Chignolin.** (A) Folding free energy comparison between reference model (blue), experimental data (yellow), and guided model (red). (B) Free energy profiles as a function of N- to C-terminal $\text{C}^\alpha$ distance. (C) Ensemble of 50 generated protein structures colored by their energy.

### 4.1.3 BIOEMU: HOMEODOMAIN

Finally, we showcase our approach on the fast-folding homeodomain EnHD HTH fragment (44 residues) using the BioEmu cgBE Lewis et al. (2024), a canonical model extensively studied experimentally (Religa et al., 2007; Religa, 2008) and computationally (Mayor et al., 2000; Lewis et al., 2024).

**Experimental Setup.** We generate ensembles with BioEmu and compare expected $^3J_{\text{HN-HA}}$ couplings to experiment. These couplings report on backbone dihedrals and thus thermodynamic populations. In our experiments, we use the 10 most informative observables (details in Appendix B.3.3). We quantified the agreement between experiments and computational predictions by the Q-factor Bax (2003) and notice that the unguided model shows a clear discrepancy ($Q = 0.147$), motivating the use of our approach. Training details are in Appendix B.3.3.

**Evaluation.** MEW markedly improves agreement while preserving physical plausibility (Fig. 3). Using only 10 experimental expectations, $Q$ drops from $0.147$ (unguided) to $0.037$ (MEW); post-hoc reweighting achieves $0.031$ but with moderate weight degeneracy (relative ESS $= 0.255$). MEW avoids this importance-weight collapse by updating the generator directly. After guidance, points cluster near the identity within experimental uncertainty; the inset ensemble shows broad conformational coverage without mode collapse, and a representative observable (residue 38) shifts toward the experimental region without variance loss. Full histograms and structural diagnostics are in Figs. 12 and 13. Overall, MEW uses sparse 1D NMR readouts to make targeted, physically consistent adjustments to BioEmu's EnHD sampling.

These results demonstrate that guidance in the sparse-data regime with MEW regularization allows us to effectively align high-dimensional and highly structured generative models with experimental constraints without compromising local physical validity and maintaining global distributional properties such as multi-modality.

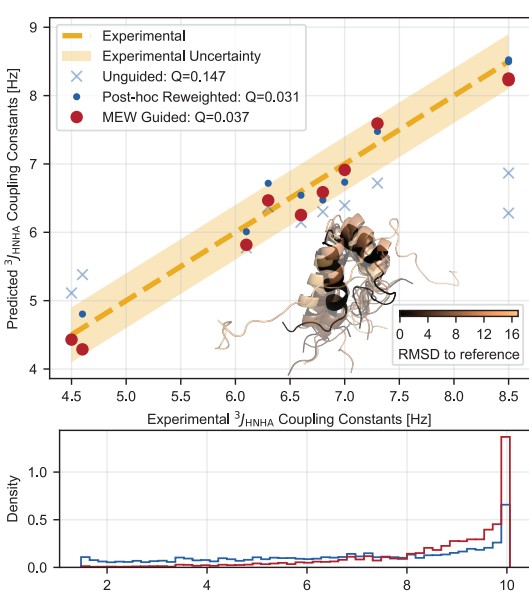

Figure 3: **Observable guidance on EnHD.** Top: 10 experimental $^3J$ vs. unguided (light blue), MEW-guided (red), and post-hoc reweighted predictions; inset: 10 sampled structures colored by RMSD. Bottom: representative histogram of residue 38 illustrating the guided population shift.

### 4.2 PATH GUIDANCE

We now use the cgBE to evaluate path guidance with MEW regularization for up-sampling high-energy transition configurations (states), which are critical for understanding the folding process of proteins. Due to their high energy (Eq. (5)), these states account for only 1% of both the data and model distribution, making their successful up-sampling a strong demonstration of our method's effectiveness. Consistent with Section 4.1, we also use the chignolin mini-protein, as its transition regions are low-probability but coherent regions, as the transition path is approximately concentrated near a dominant pathway. This geometry satisfies the precondition for effective path guidance as aggregated kernels bring a consistent pull towards $A$, whereas proteins with highly multi-modal transition states would yield competing signals and weaker guidance. To contextualize path guidance, we first introduce an alternative baseline.

**Baseline.** As a natural alternative to path guidance, we adapt loss guidance to our setting by using the log-likelihood of a KDE fitted on guiding points $\mathcal{X}_0^g = \{\mathbf{x}_0^i\}_{i=1}^M$. Specifically, we will change the perturbation kernel from Eq. (19) to $\mathcal{K}_{h_t(\varphi)}(\hat{\mathbf{x}}_t(\mathbf{x}_t), \mathcal{X}_0^g)$. While it appears similar to path guidance, the key difference lies in the space the KDE is computed. In path guidance, the kernel is applied along the trajectory $\{\mathcal{X}_t^g\}_{t=0}^{t=1}$, resulting in a distinct KDE for each time step $t$. In contrast, loss guidance computes the KDE in data space and estimates the likelihood with respect to the posterior mean

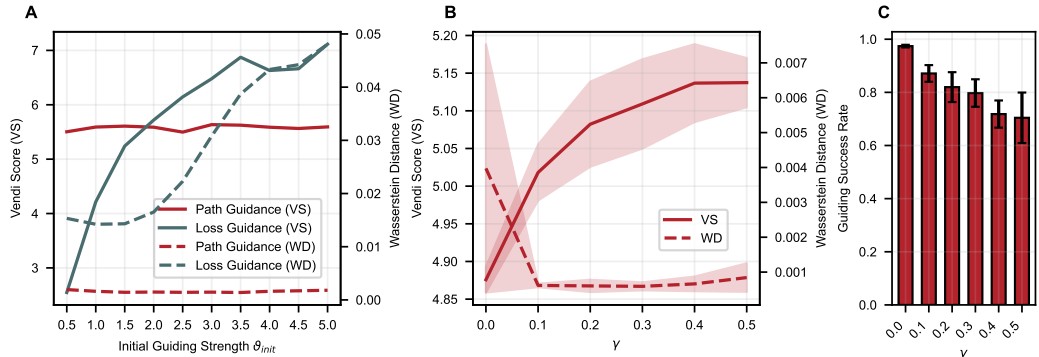

Figure 4: **Path Guidance vs. Loss-Guidance for sampling Transition States**. (A) Sample quality and diversity, measured by the Wasserstein Distance (WD) and Vendi score (VS), show that path guidance preserves diversity and quality even at high guiding strengths, whereas loss guidance deteriorates. (B) Without MEW regularization ($\gamma = 0$), sampled transition states tend to collapse and have no diversity (VS). Regularization also improves sample quality (WD). (C) Guiding success rate, measured as the percentage of transition states sampled, for different regularization strengths.

$\hat{\mathbf{x}}_t(\mathbf{x}_t)$, which requires backpropagating through the model at every sampling step. Implementation details and ablation studies for two alternative baselines that do not augment the vector field are provided in Appendix B.4 and Appendix D.3. To explore the dynamics of both methods, we design a synthetic example to study the effect of different parameters (see Appendix D.1 for details).

**Evaluation Criteria.** We assess the methods using three key metrics. First, we measure guiding success as the percentage of sampled transition configurations (see Appendix B.2 for details). Second, we evaluate the diversity among transition states using the Vendi score (VS) (Friedman & Dieng, 2022) to verify that our method generates novel samples rather than merely resampling the guiding data. Lastly, since we cannot evaluate the energy under the coarse-grained model, we instead ensure physical validity of the generated samples under guidance by computing the Wasserstein distance (WD) between the bond length distributions of generated and ground truth samples, which quantifies how well our method preserves the local molecular structure.

**Transition State Sampling.** For the transition configuration sampling task, we adapt the kernel to handle rigid-body transformations using the Kabsch algorithm (Kabsch, 1976) akin to that adopted in Pasarkar et al. (2023). Since we found loss guidance to be difficult to optimize in this application, we first performed a large grid search to identify optimal parameters for a fair comparison. This analysis revealed that increasing the guidance strength deteriorates sample quality in loss guidance, preventing it from achieving meaningful guiding success (Fig. 17B). The performance gap stems from two key disadvantages of loss guidance. First, it requires using the posterior mean to compute the augmentation factor, which, especially at large $t$, suffers from very high variance. Second, at small $t$, while the predictions become more accurate, the KDE fails to capture the distribution of the guiding points, as it is not well-suited for high data complexity. As a result, the loss signal can degrade the sampled data, as evident from the increasing Wasserstein distance as the guiding strength $\vartheta_{init}$ increases (Fig. 4A). In contrast, path guidance circumvents this issue by applying stronger guidance for larger $t$, where the latent is primarily noise, and decreasing it while sampling. Notably, in Fig. 4A, we observe that both quality and diversity remain largely unaffected by the initial guiding strength $\vartheta_{init}$. We further investigate the difference between path and loss guidance in Appendix D.2.

After observing that loss guidance could not be reliably optimized, we conducted a separate set of experiments to evaluate path guidance within the MEW framework by optimizing the objective in Eq. (14). Disabling regularization ($\gamma = 0$) results in the highest guidance success rates (Fig. 4C), but produces highly degenerate samples and reduced structural diversity, as indicated by the large variance in Wasserstein distance. In contrast, applying MEW regularization improves both sample quality and diversity (Fig. 4B), while incurring only a modest reduction in guidance success. Overall, our results demonstrate that path guidance offers a strong alternative to loss guidance, and that MEW regularization is essential for robust and physically meaningful sampling in data-sparse regimes.

## 5 RELATED WORKS

**Stochastic Optimal Control.** MEW also naturally connects to recent advances in stochastic optimal control (SOC) applied to diffusion and flow-based generative models. In particular, to approaches which consider steering generative trajectories by balancing a task-specific objective, such as aligning with experimental observables or reward models with a regularization that penalizes deviation from a pre-trained base model. In those works (Uehara et al., 2024; Domingo-Enrich et al., 2024; Han et al., 2024; Tang, 2024), fine-tuning the diffusion model is framed as a SOC problem that minimizes control effort while achieving alignment with downstream goals. Conceptually, the MEW principle plays an analogous role to the control cost in SOC, regularizing path perturbations to preserve the prior's structure while achieving target objectives. This connection puts MEW within the broader trend of leveraging control-theoretic principles, including KL and f-divergence regularization, to derive principled, sample-efficient, and robust fine-tuning strategies for probabilistic generative models.

**Transition ensemble sampling.** Traditional methods like transition path sampling (Bolhuis et al., 2000; Cabriolu et al., 2017) use Monte Carlo in trajectory space, while recent machine learning approaches (Liu et al., 2025) employ neural networks but require extensive training data or predefined collective variables. Instead of explicit path sampling, we guide the generative process using latent representations of known transition states. While related to recent work using Boltzmann Generators (Plainer et al., 2023), our approach directly modifies the score function during sampling rather than performing MCMC moves between paths, enabling more efficient exploration of transition regions.

**Reweighting with experimental data.** Reweighting molecular dynamics simulations using experimental data has a long history in computational chemistry and biophysics. Theoretical work (Roux & Weare, 2013; Cavalli et al., 2013; Boomsma et al., 2014; Pitera & Chodera, 2012) adopted Jaynes (1957) Maximum Entropy approach to the problem, following several early experimental studies (Lindorff-Larsen et al., 2005; Dedmon et al., 2004; Cesari et al., 2018) based on replica-averaged simulations, giving a theoretical foundation for these approaches. This work was later complemented by probabilistic and Bayesian perspectives (Olsson et al., 2013; Bottaro et al., 2020; Bonomi et al., 2016; 2018), some of which specifically focused on reweighing (Hummer & Köfinger, 2015; Olsson et al., 2016; 2017; Kolloff & Olsson, 2023).

## 6 LIMITATIONS

Despite the strong empirical performance of MEW guidance across a range of scientific settings, several limitations merit consideration. These primarily stem from the assumptions underpinning the method's application, e.g., that physical observables or representative samples can be leveraged to correct expectation values or guide sampling in low-density regions. While this does not require perfect model accuracy, it does require the model to be sufficiently expressive and responsive to guidance. If key modes are absent, convergence to meaningful distributions may fail. Additionally, the current framework assumes differentiable observables and guidance targets, restricting its applicability in discrete or non-differentiable domains.

## 7 CONCLUSION

In this work, we introduced minimum-excess-work (MEW) guidance, a physics-inspired framework for regularizing the guidance of pre-trained probability flow generative models by regularizing *excess work*. Our analysis shows that this thermodynamically motivated regularization is closely connected to upper bounds on the Wasserstein distance and the KL divergence between the reference and guided distributions. We demonstrated the effectiveness of MEW regularization in two settings: *Observable Guidance* and *Path Guidance*. These approaches enable alignment with sparse experimental constraints and targeted sampling in low-density regions, while maintaining model flexibility. By penalizing excess work, our method reduces bias and enhances the sampling of rare, physically meaningful configurations, without degrading sample quality. Our results position MEW guidance as a principled and effective tool for bias correction and informed exploration in data-scarce scientific applications, such as the refinement of coarse-grained force fields against experimental data or the generation of starting conditions for unbiased MD or transition path sampling.

## REPRODUCIBILITY STATEMENT

To ensure reproducibility, we release anonymised code[2] with a detailed README and scripts to reproduce all experiments under the settings reported in this work. The repository includes the pretrained checkpoints used for sampling. For path guidance we additionally provide the guiding samples, and for observable guidance, the summary statistics required to recreate the results. Implementation details, such as architectures, training schedules, and hyperparameter search procedures, are documented in Appendix B, together with compute and runtime specifications. Although the code for the synthetic examples is not included, we provide a complete description of their setup in Appendix B to enable independent reimplementation.

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

## DISCLOSURE OF LLM USAGE

For this submission, large language models were used for polishing text (improving clarity, precision, and flow) and for refinements of plots to improve visual quality and presentation.

## A PROOFS

### A.1 SHORT DERIVATION OF MAXIMUM ENTROPY REWEIGHTING OF MD TRAJECTORIES USING OBSERVABLES

The maximum entropy approach (Jaynes, 1957) has been widely adopted (Hummer & Köfinger, 2015; Boomsma et al., 2014; Olsson et al., 2016; 2017; Bottaro et al., 2020) to derive reweighting schemes to find a minimally biased probability distribution that satisfies experimental constraints.

Consider a reference probability distribution $p(\mathbf{x})$, e.g., an empirical distribution estimated from MD simulation data, and an unknown target distribution $p'(\mathbf{x})$ that should match experimental measurements. Following Jaynes' maximum entropy principle, we seek to minimize the KL divergence from $p(\mathbf{x})$ to $p'(\mathbf{x})$ subject to the constraint that the expectations of observables $O_i(\mathbf{x})$ under $p'(\mathbf{x})$ match their experimental values $o_i$. That is,

$$\min_{p'} \int p'(\mathbf{x}) \log \frac{p'(\mathbf{x})}{p(\mathbf{x})} \, \mathrm{d}\mathbf{x} \tag{20}$$

subject to:

$$\mathbb{E}_{p'(\mathbf{x})}[O_i(\mathbf{x})] = o_i \ \text{ for } i = 1, \dots, M \tag{21}$$

$$\int p'(\mathbf{x}) \, \mathrm{d}\mathbf{x} = 1 \tag{22}$$

Using the method of Lagrange multipliers, we obtain the following objective:

$$S = -\int p'(\mathbf{x}) \log \frac{p'(\mathbf{x})}{p(\mathbf{x})} \, \mathrm{d}\mathbf{x} + \sum_{i=1}^{M} \lambda_i \left( \int p'(\mathbf{x}) O_i(\mathbf{x}) \, \mathrm{d}\mathbf{x} - o_i \right) + \mu \left( \int p'(\mathbf{x}) \, \mathrm{d}\mathbf{x} - 1 \right) \tag{23}$$

where $\{\lambda_i\}_{i=1}^{M}$ are the Lagrange multipliers for the constraints on the $M$ observables, and $\mu$ is the multiplier for density normalization. Setting the functional derivative $\delta S / \delta p'$ to zero yields

$$-\log \frac{p'(\mathbf{x})}{p(\mathbf{x})} - 1 + \sum_{i=1}^{M} \lambda_i O_i(\mathbf{x}) + \mu = 0 \,. \tag{24}$$

Finally, solving for $p'(\mathbf{x})$ and determining $\mu$ through normalization gives

$$p'(\mathbf{x}) \propto p(\mathbf{x}) \exp\left( -\sum_{i=1}^{M} \lambda_i O_i(\mathbf{x}) \right) \,, \tag{25}$$

where the $\lambda$s are determined, e.g., following Bottaro et al. (2020), such that the constraints on the expectations are satisfied. This reweighted distribution represents the maximum entropy solution that satisfies the experimental constraints while minimizing the bias introduced relative to the reference distribution $p(\mathbf{x})$.

### A.2 BOUNDING THE WASSERSTEIN DISTANCE

In this section, we derive an upper bound on the squared Wasserstein distance $W_2^2(p_0, p_0')$, where the distributions $p_0$ and $p_0'$ are obtained by evolving a common terminal distribution $p_1 = p_1'$ backward in time according to the ODEs in Eqs. (4) and (9). We begin by proving a Grönwall-type lemma (see, e.g., Bressan & Piccoli (2007, Lemma 2.1.2)) that will be useful to prove our result.

**Lemma A.1.** *Let $T > 0$ and let $f$ be an absolutely continuous function over $[0, T]$ satisfying the differential inequality*

$$\frac{\mathrm{d}}{\mathrm{d}t} f(t) \leq a(t) f(t) + b(t) \qquad \text{for a.e. } t \in [0, T], \tag{26}$$

*where $a, b \in L^1([0, T])$ are integrable functions. Then, for every $t \in [0, T]$,*

$$f(t) \leq \exp\left(\int_0^t a(u)\,\mathrm{d}u\right) f(0) + \int_0^t \exp\left(\int_s^t a(u)\,\mathrm{d}u\right) b(s)\,\mathrm{d}s . \tag{27}$$

*Proof.* Define the absolutely continuous function

$$\psi(t) := \exp\left(-\int_0^t a(u)\,\mathrm{d}u\right)$$

and note that $\psi(t) > 0$, $\psi(0) = 1$, and

$$\frac{\mathrm{d}}{\mathrm{d}t}\psi(t) = -a(t)\psi(t) .$$

Multiplying both sides of Eq. (26) by $\psi(t)$ and integrating from 0 to $t$, we have

$$\int_0^t \psi(s)\frac{\mathrm{d}}{\mathrm{d}s}f(s)\,\mathrm{d}s \leq \int_0^t \psi(s)a(s)f(s)\,\mathrm{d}s + \int_0^t \psi(s)b(s)\,\mathrm{d}s \tag{28}$$

$$\psi(t)f(t) - \psi(0)f(0) - \int_0^t \psi'(s)f(s)\,\mathrm{d}s \leq \int_0^t \psi(s)a(s)f(s)\,\mathrm{d}s + \int_0^t \psi(s)b(s)\,\mathrm{d}s \tag{29}$$

$$\psi(t)f(t) - f(0) + \int_0^t a(s)\psi(s)f(s)\,\mathrm{d}s \leq \int_0^t \psi(s)a(s)f(s)\,\mathrm{d}s + \int_0^t \psi(s)b(s)\,\mathrm{d}s \tag{30}$$

$$\psi(t)f(t) \leq f(0) + \int_0^t \psi(s)b(s)\,\mathrm{d}s . \tag{31}$$

We then divide both sides by $\psi(t)$ again to conclude:

$$f(t) \leq \frac{f(0)}{\psi(t)} + \int_0^t \frac{\psi(s)}{\psi(t)}b(s)\,\mathrm{d}s \tag{32}$$

$$= \exp\left(\int_0^t a(u)\,\mathrm{d}u\right) f(0) + \int_0^t \exp\left(\int_s^t a(u)\,\mathrm{d}u\right) b(s)\,\mathrm{d}s \tag{33}$$

$\square$

**Proposition A.2.** *Let $T > 0$, and let $\mathbf{v}, \mathbf{v}' : [0, T] \times \mathbb{R}^d \to \mathbb{R}^d$ be measurable in time and $L_t$-Lipschitz in space, with $L_t$ integrable. Let $p_0$ be a probability measure on $\mathbb{R}^d$, and define $p_t, p_t'$ as the pushforwards of $p_0$ under the flows of the ODEs $\frac{\mathrm{d}\mathbf{x}_t}{\mathrm{d}t} = \mathbf{v}_t(\mathbf{x}_t)$ and $\frac{\mathrm{d}\mathbf{x}_t'}{\mathrm{d}t} = \mathbf{v}_t'(\mathbf{x}_t')$. Then for all $t \in [0, T]$,*

$$W_2^2(p_t, p_t') \leq \int_0^t \exp\left(t - s + 2\int_s^t L_u\,\mathrm{d}u\right) \mathbb{E}_{\mathbf{x}\sim p_s'}\left[\|\mathbf{v}_s(\mathbf{x}) - \mathbf{v}_s'(\mathbf{x})\|^2\right]\,\mathrm{d}s . \tag{34}$$

*Proof.* Let $\phi_t, \phi_t'$ be the flows of the ODEs, i.e., $\mathbf{x}_t = \phi_t(\mathbf{x}_0)$, $\frac{\mathrm{d}\phi_t(\mathbf{x})}{\mathrm{d}t} = \mathbf{v}_t(\phi_t(\mathbf{x}))$, and similarly for $\mathbf{x}'$ and $\phi_t'$. Define the coupling:

$$\tilde{\pi}_t := (\phi_t, \phi_t')_* p_0 \in \Gamma(p_t, p_t') , \tag{35}$$

i.e., the pushforward of $p_0$ through the map $\mathbf{x} \mapsto (\phi_t(\mathbf{x}), \phi_t'(\mathbf{x}))$. By definition of 2-Wasserstein distance, we can write:

$$W_2^2(p_t, p_t') \leq \int \|\mathbf{x} - \mathbf{x}'\|^2\,\mathrm{d}\tilde{\pi}_t(\mathbf{x}, \mathbf{x}') = \mathbb{E}_{(\mathbf{x}_t, \mathbf{x}_t')\sim\tilde{\pi}_t}\left[\|\mathbf{x}_t - \mathbf{x}_t'\|^2\right] \tag{36}$$

Take any $\mathbf{x}_0, \mathbf{x}_0' \in \mathbb{R}^d$ and let $\mathbf{x}_t = \phi_t(\mathbf{x}_0)$ and $\mathbf{x}_t' = \phi_t'(\mathbf{x}_0')$. Then,

$$\frac{\mathrm{d}}{\mathrm{d}t}\|\mathbf{x}_t - \mathbf{x}_t'\|^2 = 2(\mathbf{x}_t - \mathbf{x}_t') \cdot (\mathbf{v}_t(\mathbf{x}_t) - \mathbf{v}_t'(\mathbf{x}_t')) \tag{37}$$

$$= 2(\mathbf{x}_t - \mathbf{x}_t') \cdot (\mathbf{v}_t(\mathbf{x}_t) - \mathbf{v}_t(\mathbf{x}_t')) + 2(\mathbf{x}_t - \mathbf{x}_t') \cdot (\mathbf{v}_t(\mathbf{x}_t') - \mathbf{v}_t'(\mathbf{x}_t')) \tag{38}$$

We bound the first term using the Cauchy–Schwarz inequality and the $L_t$-Lipschitzness of $\mathbf{v}_t$:

$$2(\mathbf{x}_t - \mathbf{x}'_t) \cdot (\mathbf{v}_t(\mathbf{x}_t) - \mathbf{v}_t(\mathbf{x}'_t)) \leq 2\|\mathbf{x}_t - \mathbf{x}'_t\| \, \|\mathbf{v}_t(\mathbf{x}_t) - \mathbf{v}_t(\mathbf{x}'_t)\| \tag{39}$$

$$\leq 2L_t\|\mathbf{x}_t - \mathbf{x}'_t\|^2 . \tag{40}$$

Using $0 \leq \|\mathbf{a} - \mathbf{b}\|^2 = \|\mathbf{a}\|^2 + \|\mathbf{b}\|^2 - 2\mathbf{a} \cdot \mathbf{b}$ for the second term we have:

$$2(\mathbf{x}_t - \mathbf{x}'_t) \cdot (\mathbf{v}_t(\mathbf{x}'_t) - \mathbf{v}'_t(\mathbf{x}'_t)) \leq \|\mathbf{x}_t - \mathbf{x}'_t\|^2 + \|\mathbf{v}_t(\mathbf{x}'_t) - \mathbf{v}'_t(\mathbf{x}'_t)\|^2 . \tag{41}$$

Plugging these two bounds into Eq. (38), we get

$$\frac{\mathrm{d}}{\mathrm{d}t}\|\mathbf{x}_t - \mathbf{x}'_t\|^2 = 2(\mathbf{x}_t - \mathbf{x}'_t) \cdot (\mathbf{v}_t(\mathbf{x}_t) - \mathbf{v}_t(\mathbf{x}'_t)) + 2(\mathbf{x}_t - \mathbf{x}'_t) \cdot (\mathbf{v}_t(\mathbf{x}'_t) - \mathbf{v}'_t(\mathbf{x}'_t)) \tag{42}$$

$$\leq 2L_t\|\mathbf{x}_t - \mathbf{x}'_t\|^2 + \|\mathbf{x}_t - \mathbf{x}'_t\|^2 + \|\mathbf{v}_t(\mathbf{x}'_t) - \mathbf{v}'_t(\mathbf{x}'_t)\|^2 \tag{43}$$

$$= (2L_t + 1)\|\mathbf{x}_t - \mathbf{x}'_t\|^2 + \|\mathbf{v}_t(\mathbf{x}'_t) - \mathbf{v}'_t(\mathbf{x}'_t)\|^2 . \tag{44}$$

Finally, taking expectations on both sides w.r.t. $(\mathbf{x}_t, \mathbf{x}'_t) \sim \tilde{\pi}_t$, and exchanging expectation and derivative under standard regularity assumptions, we get:

$$\frac{\mathrm{d}}{\mathrm{d}t}\mathbb{E}\left[\|\mathbf{x}_t - \mathbf{x}'_t\|^2\right] \leq (2L_t + 1)\,\mathbb{E}\left[\|\mathbf{x}_t - \mathbf{x}'_t\|^2\right] + \mathbb{E}\left[\|\mathbf{v}_t(\mathbf{x}'_t) - \mathbf{v}'_t(\mathbf{x}'_t)\|^2\right] . \tag{45}$$

This inequality can be expressed as

$$\frac{\mathrm{d}f(t)}{\mathrm{d}t} \leq (2L_t + 1)f(t) + b(t) , \qquad f(0) = 0 , \tag{46}$$

with

$$f(t) := \mathbb{E}_{(\mathbf{x}_t, \mathbf{x}'_t) \sim \tilde{\pi}_t}\left[\|\mathbf{x}_t - \mathbf{x}'_t\|^2\right] \tag{47}$$

$$b(t) := \mathbb{E}_{\mathbf{x}'_t \sim p'_t}\left[\|\mathbf{v}_t(\mathbf{x}'_t) - \mathbf{v}'_t(\mathbf{x}'_t)\|^2\right] . \tag{48}$$

Applying Lemma A.1 with $a(t) = (2L_t + 1)$, we get:

$$f(t) \leq \int_0^t \exp\left(\int_s^t (2L_u + 1)\,\mathrm{d}u\right) b(s)\,\mathrm{d}s \tag{49}$$

$$= \int_0^t e^{t-s} \exp\left(2\int_s^t L_u\,\mathrm{d}u\right) b(s)\,\mathrm{d}s \tag{50}$$

Since from Eq. (36) we know that $W_2^2(p_t, p'_t) \leq f(t)$, the statement follows:

$$W_2^2(p_t, p'_t) \leq \int_0^t e^{t-s} \exp\left(2\int_s^t L_u\,\mathrm{d}u\right) \mathbb{E}_{\mathbf{x} \sim p'_s}\left[\|\mathbf{v}_s(\mathbf{x}) - \mathbf{v}'_s(\mathbf{x})\|^2\right]\,\mathrm{d}s . \tag{51}$$

$$\square$$

Although the result in the time-reversed case is straightforward as it directly follows from a time reparameterization, we state it and prove it for the sake of completeness.

**Proposition A.3.** *Let* $\mathbf{v}, \mathbf{v}' : [0,1] \times \mathbb{R}^d \to \mathbb{R}^d$ *be measurable in time and* $L_t$*-Lipschitz in space, with* $L_t$ *integrable. Let* $p_0, p'_0$ *be probability measures on* $\mathbb{R}^d$*, and define* $p_t, p'_t$ *as the pushforwards of* $p_0$ *under the flows of the ODEs* $\frac{\mathrm{d}\mathbf{x}_t}{\mathrm{d}t} = \mathbf{v}_t(\mathbf{x}_t)$ *and* $\frac{\mathrm{d}\mathbf{x}'_t}{\mathrm{d}t} = \mathbf{v}'_t(\mathbf{x}'_t)$*. Assume* $p_1 = p'_1$*. Then,*

$$W_2^2(p_0, p'_0) \leq \int_0^1 \exp\left(t + 2\int_0^t L_s\,\mathrm{d}s\right) \mathbb{E}_{\mathbf{x} \sim p'_t}\left[\|\mathbf{v}_t(\mathbf{x}) - \mathbf{v}'_t(\mathbf{x})\|^2\right]\,\mathrm{d}t . \tag{52}$$

*Proof.* Consider the time reversal transformation $s = 1 - t$. Define $\tilde{\mathbf{x}}_s := \mathbf{x}_{1-s}$ and $\tilde{\mathbf{x}}'_s := \mathbf{x}'_{1-s}$, where $\mathbf{x}_t$ and $\mathbf{x}'_t$ satisfy the original ODEs with vector fields $\mathbf{v}_t, \mathbf{v}'_t$, with $\mathbf{x}_t \sim p_t$, $\mathbf{x}'_t \sim p'_t$, and $p_1 = p'_1$. Differentiating the reversed processes, we get:

$$\frac{\mathrm{d}\tilde{\mathbf{x}}_s}{\mathrm{d}s} = \frac{\mathrm{d}\mathbf{x}_{1-s}}{\mathrm{d}t} \cdot \frac{\mathrm{d}t}{\mathrm{d}s} = -\mathbf{v}_{1-s}(\mathbf{x}_{1-s}) = -\mathbf{v}_{1-s}(\tilde{\mathbf{x}}_s) \tag{53}$$

and similarly for $\tilde{\mathbf{x}}'$. Thus, the reversed processes satisfy:

$$\frac{\mathrm{d}\tilde{\mathbf{x}}_s}{\mathrm{d}s} = \tilde{\mathbf{v}}_s(\tilde{\mathbf{x}}_s) , \qquad \frac{\mathrm{d}\tilde{\mathbf{x}}'_s}{\mathrm{d}s} = \tilde{\mathbf{v}}'_s(\tilde{\mathbf{x}}'_s) , \qquad (54)$$

where we defined the reversed velocity fields $\tilde{\mathbf{v}}_s(\mathbf{x}) := -\mathbf{v}_{1-s}(\mathbf{x})$ and $\tilde{\mathbf{v}}'_s(\mathbf{x}) := -\mathbf{v}'_{1-s}(\mathbf{x})$. From the definitions $\tilde{\mathbf{x}}_s := \mathbf{x}_{1-s}$ and $\tilde{\mathbf{x}}'_s := \mathbf{x}'_{1-s}$ it directly follows that $\tilde{p}_s = p_{1-s}$ and $\tilde{p}'_s = p'_{1-s}$. At $s = 0$, we have $\tilde{p}_0 = p_1 = p'_1 = \tilde{p}'_0$, so the reversed processes start from the same distribution.

Since $\mathbf{v}_t$ and $\mathbf{v}'_t$ are $L_t$-Lipschitz in space with $L_t$ integrable, $\tilde{\mathbf{v}}_s$ and $\tilde{\mathbf{v}}'_s$ are $L_{1-s}$-Lipschitz. The reversed ODEs start at $s = 0$ from the same distribution ($\tilde{p}_0 = \tilde{p}'_0$) and evolve to $\tilde{p}_1 = p_0$ and $\tilde{p}'_1 = p'_0$ at $s = 1$. Applying Proposition A.2, we get:

$$W_2^2(\tilde{p}_1, \tilde{p}'_1) \leq \int_0^1 \exp\left(1 - s + 2\int_s^1 L_{1-u}\,\mathrm{d}u\right) \mathbb{E}_{\mathbf{x}\sim\tilde{p}'_s}\left[\|\tilde{\mathbf{v}}_s(\mathbf{x}) - \tilde{\mathbf{v}}'_s(\mathbf{x})\|^2\right]\,\mathrm{d}s . \qquad (55)$$

Substituting $\tilde{p}_s = p_{1-s}$ and $\tilde{p}'_s = p'_{1-s}$, using the definitions of $\tilde{\mathbf{v}}_t, \tilde{\mathbf{v}}'_t$, and applying a change of variables $t = 1 - s$, we obtain the desired bound:

$$W_2^2(p_0, p'_0) \leq \int_0^1 \exp\left(1 - s + 2\int_s^1 L_{1-u}\,\mathrm{d}u\right) \mathbb{E}_{\mathbf{x}\sim p'_{1-s}}\left[\|\mathbf{v}_{1-s}(\mathbf{x}) - \mathbf{v}'_{1-s}(\mathbf{x})\|^2\right]\,\mathrm{d}s \qquad (56)$$

$$= \int_0^1 \exp\left(t + 2\int_{1-t}^1 L_{1-u}\,\mathrm{d}u\right) \mathbb{E}_{\mathbf{x}\sim p'_t}\left[\|\mathbf{v}_t(\mathbf{x}) - \mathbf{v}'_t(\mathbf{x})\|^2\right]\,\mathrm{d}t \qquad (57)$$

$$= \int_0^1 \exp\left(t + 2\int_0^t L_s\,\mathrm{d}s\right) \mathbb{E}_{\mathbf{x}\sim p'_t}\left[\|\mathbf{v}_t(\mathbf{x}) - \mathbf{v}'_t(\mathbf{x})\|^2\right]\,\mathrm{d}t . \qquad (58)$$

$$\square$$

In this work, we are specifically interested in the ODEs (4) and (9):

**Proposition 3.1.** *Let $p_t$ and $p'_t$ be the distributions at time $t$ obtained by solving the ODEs (4) and (9) backwards in time from the same initial distribution $p_1$ at $t = 1$. Assume that the vector fields are measurable in time and $L_t$-Lipschitz in space with $L_t$ integrable. Then:*

$$W_2^2(p_0, p'_0) \leq \int_0^1 w_{\mathrm{W}}(t)\,\frac{g(t)^4}{4}\,\mathbb{E}_{\mathbf{x}\sim p'_t}\left[\|\mathbf{h}_\vartheta(\mathbf{x}, t)\|^2\right]\,\mathrm{d}t , \qquad w_{\mathrm{W}}(t) := e^{t + 2\int_0^t L_s\,\mathrm{d}s} . \qquad (12)$$

*Proof.* The ODEs (4) and (9) have the following vector fields:

$$\mathbf{v}_t(\mathbf{x}) = \mathbf{f}(\mathbf{x}, t) - \frac{1}{2}g(t)^2 \mathbf{s}(\mathbf{x}, t)$$

$$\mathbf{v}'_t(\mathbf{x}) = \mathbf{f}(\mathbf{x}, t) - \frac{1}{2}g(t)^2 \left(\mathbf{s}(\mathbf{x}, t) + \mathbf{h}(\mathbf{x}, t)\right) .$$

The result directly follows by applying Proposition A.3:

$$W_2^2(p_0, p'_0) \leq \int_0^1 \exp\left(t + 2\int_0^t L_s\,\mathrm{d}s\right) \frac{g(t)^4}{4}\mathbb{E}_{\mathbf{x}\sim p'_t}\left[\|\mathbf{h}_\vartheta(\mathbf{x}, t)\|^2\right]\,\mathrm{d}t . \qquad (59)$$

$$\square$$

## A.3 BOUNDING THE KL DIVERGENCE

**Proposition A.4.** *Let $p, p' : \mathbb{R}^d \times [0, 1] \to \mathbb{R}_{\geq 0}$ be two probability paths over time $t \in [0, 1]$, induced by two reverse-time SDEs:*

$$\mathrm{d}\mathbf{x}_t = \boldsymbol{\mu}_t(\mathbf{x}_t)\,\mathrm{d}t + g_t\,\mathrm{d}\widetilde{\mathbf{w}}_t , \qquad \mathrm{d}\mathbf{x}_t = \boldsymbol{\mu}'_t(\mathbf{x}_t)\,\mathrm{d}t + g_t\,\mathrm{d}\widetilde{\mathbf{w}}'_t \qquad (60)$$

*where $\widetilde{\mathbf{w}}_t, \widetilde{\mathbf{w}}'_t$ are reverse-time Wiener processes, $\boldsymbol{\mu}, \boldsymbol{\mu}' : \mathbb{R}^d \times [0, 1] \to \mathbb{R}^d$, and $g : [0, 1] \to \mathbb{R}_{>0}$. Assume that $p_1 = p'_1$, that both SDEs admit strong solutions, and that $\mathbb{P}' \ll \mathbb{P}$, where $\mathbb{P}, \mathbb{P}'$ are the path measures induced by the SDEs on $C([0, 1], \mathbb{R}^d)$. Then:*

$$D_{\mathrm{KL}}(p'_0 \| p_0) \leq \frac{1}{2}\int_0^1 \frac{1}{g_t^2}\,\mathbb{E}_{\mathbf{x}\sim p'_t}\left[\|\boldsymbol{\mu}'_t(\mathbf{x}) - \boldsymbol{\mu}_t(\mathbf{x})\|^2\right]\,\mathrm{d}t . \qquad (61)$$

*Proof.* By applying the chain rule of the KL divergence (Léonard, 2014, Theorem 2.4) at $t = 0$ and $t = 1$, we have:

$$D_{\mathrm{KL}}(\mathbb{P}'\|\mathbb{P}) = D_{\mathrm{KL}}(p_0'\|p_0) + \mathbb{E}_{\mathbf{x}_0^* \sim p_0'}\Big[ \underbrace{D_{\mathrm{KL}}(\mathbb{P}'_{\mathbf{x}_0=\mathbf{x}_0^*}\|\mathbb{P}_{\mathbf{x}_0=\mathbf{x}_0^*})}_{\geq 0} \Big] \tag{62}$$

$$D_{\mathrm{KL}}(\mathbb{P}'\|\mathbb{P}) = \underbrace{D_{\mathrm{KL}}(p_1'\|p_1)}_{=0} + \mathbb{E}_{\mathbf{x}_1^* \sim p_1'}\Big[ D_{\mathrm{KL}}(\mathbb{P}'_{\mathbf{x}_1=\mathbf{x}_1^*}\|\mathbb{P}_{\mathbf{x}_1=\mathbf{x}_1^*}) \Big] . \tag{63}$$

The subscripts on the path measures denote conditioning on the value of the process at a specific time (by disintegration of path measures). We can therefore bound $D_{\mathrm{KL}}(p_0'\|p_0)$ by a KL divergence between path measures:

$$D_{\mathrm{KL}}(p_0'\|p_0) \leq \mathbb{E}_{\mathbf{x}_1^* \sim p_1'}\Big[ D_{\mathrm{KL}}(\mathbb{P}'_{\mathbf{x}_1=\mathbf{x}_1^*}\|\mathbb{P}_{\mathbf{x}_1=\mathbf{x}_1^*}) \Big] . \tag{64}$$

By Girsanov's theorem (Øksendal, 2003),

$$D_{\mathrm{KL}}(\mathbb{P}'_{\mathbf{x}_1=\mathbf{x}_1^*}\|\mathbb{P}_{\mathbf{x}_1=\mathbf{x}_1^*}) = \frac{1}{2}\mathbb{E}_{\mathbb{P}'_{\mathbf{x}_1=\mathbf{x}_1^*}}\left[ \int_0^1 \frac{1}{g_t^2}\|\boldsymbol{\mu}_t'(\mathbf{x}_t) - \boldsymbol{\mu}_t(\mathbf{x}_t)\|^2\, \mathrm{d}t \right] . \tag{65}$$

We can now write the iterated expectation as an expectation over the unconditional path measure $\mathbb{P}'$:

$$\mathbb{E}_{\mathbf{x}_1^* \sim p_1'}\Big[ D_{\mathrm{KL}}(\mathbb{P}'_{\mathbf{x}_1=\mathbf{x}_1^*}\|\mathbb{P}_{\mathbf{x}_1=\mathbf{x}_1^*}) \Big] = \frac{1}{2}\mathbb{E}_{\mathbf{x}_1^* \sim p_1'}\left[ \mathbb{E}_{\mathbb{P}'_{\mathbf{x}_1=\mathbf{x}_1^*}}\left[ \int_0^1 \frac{1}{g_t^2}\|\boldsymbol{\mu}_t'(\mathbf{x}_t) - \boldsymbol{\mu}_t(\mathbf{x}_t)\|^2\, \mathrm{d}t \right]\right] \tag{66}$$

$$= \frac{1}{2}\mathbb{E}_{\mathbb{P}'}\left[ \int_0^1 \frac{1}{g_t^2}\|\boldsymbol{\mu}_t'(\mathbf{x}_t) - \boldsymbol{\mu}_t(\mathbf{x}_t)\|^2\, \mathrm{d}t \right] . \tag{67}$$

Finally, we switch the expectation and integral (Fubini–Tonelli), and simplify the expectation over $\mathbb{P}'$ into an expectation over the time marginal $p_t'$ since the argument of the integral only depends on $t$:

$$\mathbb{E}_{\mathbb{P}'}\left[ \int_0^1 \frac{1}{g_t^2}\|\boldsymbol{\mu}_t'(\mathbf{x}_t) - \boldsymbol{\mu}_t(\mathbf{x}_t)\|^2\, \mathrm{d}t \right] = \int_0^1 \frac{1}{g_t^2}\, \mathbb{E}_{\mathbf{x} \sim p_t'}\left[ \|\boldsymbol{\mu}_t'(\mathbf{x}) - \boldsymbol{\mu}_t(\mathbf{x})\|^2 \right]\, \mathrm{d}t , \tag{68}$$

which concludes the proof. □

In this work, we are specifically interested in the reverse-time SDEs (3) and (8):

**Proposition 3.2.** *Let $p_t$ and $p_t'$ be the distributions at time $t$ induced by the reverse-time SDEs* (3) *and* (8) *starting from the same distribution $p_1$ at $t = 1$. Assume that both SDEs admit strong solutions, and that $\mathbb{P}' \ll \mathbb{P}$, where $\mathbb{P}, \mathbb{P}'$ are the path measures induced by the SDEs on $C([0,1], \mathbb{R}^d)$. Then:*

$$D_{\mathrm{KL}}(p_0'\|p_0) \leq \int_0^1 w_{\mathrm{KL}}(t)\, \frac{g(t)^4}{4}\, \mathbb{E}_{\mathbf{x} \sim p_t'}\left[ \|\mathbf{h}_\vartheta(\mathbf{x}, t)\|^2 \right]\, \mathrm{d}t , \qquad w_{\mathrm{KL}}(t) := \frac{2}{g(t)^2} . \tag{13}$$

*Proof.* The result directly follows by applying Proposition A.4 to the drifts of the reverse-time SDEs (3) and (8). □

# B EXPERIMENTAL DETAILS

## B.1 COARSE-GRAINED BOLTZMANN EMULATOR MODEL ARCHITECTURE AND TRAINING SETUP

The score function in this work is based on the CPaiNN architecture introduced in (Schreiner et al., 2023) with $n_h = 64$ hidden features and five message passing layers. The score is calculated in two steps - embedding and processing by CPaiNN. In the embedding step, each node is embedded using a lookup function. The pairwise distances between nodes and the diffusion time $t$ is encoded with a positional embedding as described in (Vaswani, 2017). The embedded $t$ is concatenated to the node features and the resulting vector is projected down to $n_h$ dimensions using an MLP. Additionally, each node is assigned $n_h$ zero-vectors serving as initial equivariant features.

The embedded graph is processed by the score model and the final equivariant features are read out as the score.

The score model was trained in a DDPM setup as described in (Schreiner et al., 2023) using an exponential moving average (Tarvainen & Valpola, 2017) with a decay value of 0.99, batch size of 128, and the Adam optimizer with a learning rate of 0.001, $\beta_1 = .9$, $\beta_2 = .999$.

### B.2 ANALYSIS OF CLN025 MD TRAJECTORY

To evaluate our methods, we calculate pair-wise $C^\alpha$ distances of the ten-residue miniprotein and project those features onto the two slowest time-lagged independent components (Pérez-Hernández et al., 2013) with a lag time $\tau = 10$ ns. We then clustered the MD trajectory into $n = 128$ states using KMeans. The discretized trajectory was then used for estimating a Markov State Model (MSM) (Prinz et al., 2011; Bowman et al., 2014; Kolloff & Olsson, 2024) using a lag time of $\tau = 10$ ns (Hoffmann et al., 2021). For detailed discussions on the background and use of these methods, we refer the reader to (Bowman et al., 2009; Pande et al., 2010; Prinz et al., 2011; Husic & Pande, 2018).

#### B.2.1 COMMITTOR PROBABILITIES AND TRANSITION STATES.

In order to identify the transition states, we computed the committor probabilities (Metzner et al., 2009), defining transition states as those with values near 0.5. If we consider a reactive process of a system on a space $\Omega$ going from a state $A \subset \Omega$ to another state $B \subset \Omega$, s. t. $A \cap B = \emptyset$, the committor $q_i$ describes the probability of reaching state B before A starting from $i$.(E. & Vanden-Eijnden, 2006) Considering the protein folding process, A is the unfolded state and B is the folded state, $q_i = P(\text{folded first} \mid \text{starting at state i})$. Most importantly in our context, are states with committor values near 0.5, indicating an equal likelihood of folding or unfolding, which are identified as *transition states* (Fig. 5). These states often represent critical bottlenecks in the folding process or in chemical reactions and are thus of significant biophysical and chemical interest.

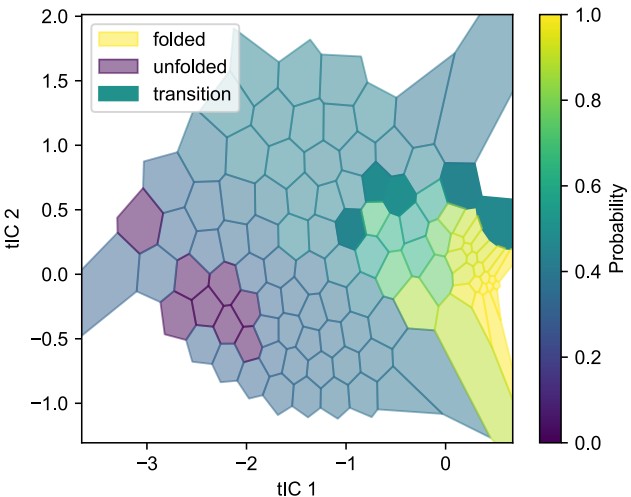

Figure 5: **Committor Probability Voronoi Diagram.** Each region is colored by its committor probability, where values near 1 correspond to folded states and values near 0 correspond to unfolded states. Regions near 0.5 represent transition states.

### B.3 OBSERVABLE GUIDANCE

We evaluated our method on two systems: a synthetic one-dimensional model and the chignolin protein system. For both systems, guidance parameters were optimized using Bayesian optimization with Gaussian Processes (GPs) implemented via scikit-optimize (Head et al., 2021). The scaling function took the form $\eta_t(\vartheta) = \eta_{\text{init}} \exp(-\kappa(1-t))$, with system-specific search spaces for $\eta_{\text{init}}$ and $\kappa$. All optimizations used 64 function evaluations with a convergence threshold of 1e-5, retaining the 5 best parameter sets. The scaling hyperparameter $\gamma$, which balances observable matching and minimum excess work, was consistently set to 1e-3 after hyperparameter search.

#### B.3.1 SYNTHETIC SETUP AND ADDITIONAL RESULTS

**Task and data.** We train a diffusion model on samples from a biased 1D quadruple-well potential (Prinz et al., 2011), allowing for direct, distribution-level validation of a system that displays multi-modality and metastability yet has a numerically accessible unbiased Boltzmann distribution.

Table 2: Gaussian Mixture Model Component Parameters

| Component | Mean ($\mu$) | Variance ($\sigma^2$) | Weight ($w$) |
|-----------|--------------|----------------------|--------------|
| 1 | 0.30 | 0.01 | 0.35 |
| 2 | -0.24 | 0.01 | 0.22 |
| 3 | 0.69 | 0.01 | 0.27 |
| 4 | -0.71 | 0.01 | 0.16 |

**Neural Network Architecture and Training.** Two multilayer perceptron (MLP) networks were trained on the Prinz potential system (Prinz et al., 2011) with $k_{\mathrm{B}} = 1.38 \cdot 10^{-23}$ and $T = 300$ K: one on the unbiased potential and another incorporating a linear bias of -4. Both networks were trained for 15,000 epochs using a batch size of 256 and the Adam optimizer with a learning rate of 1e-3. The networks shared identical architectures, with input dimension corresponding to single-atom ($n_{\mathrm{atoms}} = 1$) one-dimensional data, a time embedding dimension of 3, hidden dimension of 64, and output dimension of 1. The training process employed a linear beta scheduler with parameters $a = 0.1$ and $b = 20.0$. This scheduler controlled the noise scale during training, allowing for progressive refinement of the learned distributions.

Observable Function Parameterization. For the synthetic system, the observable function was implemented as a Gaussian Mixture Model (GMM) with four components, parameterized as shown in Table 2. The Lagrange multiplier was calculated to be -0.66 following (Bottaro et al., 2020). The parameter search space was defined as $\eta_{\mathrm{init}} \in [1.0, 20.0]$ and $\kappa \in [1.0, 20.0]$.

**Evaluation metrics.** We report (i) $\mathbb{E}_{p_{\mathcal{M}}(\mathbf{x})}[O(\mathbf{x})]$ to assess constraint satisfaction and (ii) $\mathrm{KL}(p_{\mathrm{GT}} \parallel p_{\mathcal{M}})$ to assess distributional fidelity relative to the ground truth (GT). We compare a biased reference model, the guided model, and GT.

**Main result.** As shown in Fig. 6, guidance corrects the biased density toward GT. Quantitatively (Table 3), KL is reduced by $\sim 10\times$ (from 0.13 to $0.019 \pm 0.002$) and the observable expectation moves from $-13.6$ to $11.95 \pm 0.22$, closely matching GT 12.01.

**Ablation on MEW regularization.** We compare training with ($\gamma > 0$) and without ($\gamma = 0$) MEW. Both variants match the observable expectation, but without MEW we observe mode collapse with mass concentrated in a narrow region and elevated KL, indicating poor distributional fidelity. MEW preserves the broader reference shape and stabilizes training (visuals in Fig. 14; summary metrics in Table 4).

**Conclusion.** This synthetic experiment demonstrates that observable guidance recovers the correct distribution using only expectation values, while MEW regularization prevents degenerate solutions and stabilizes training.

### B.3.2 CGBE: CHIGNOLIN

For the chignolin protein system, we defined the observable function using the interatomic distance between the first and last $\mathrm{C}^{\alpha}$ atoms ($\mathrm{C}_1^{\alpha}$ and $\mathrm{C}_{10}^{\alpha}$). The folding free energy was calculated as:

$$\Delta G = -k_B T \log \left( \frac{p_f}{1 - p_f} \right) \tag{69}$$

where $p_f$ represents the fraction of folded samples, defined using a distance cutoff of 7.5 Å. The Lagrange multiplier was determined to be -0.5 (Bottaro et al., 2020). The parameter search space was set to $\eta_{\mathrm{init}} \in [10^{-2}, 1.0]$ and $\kappa \in [1.0, 10.0]$. The optimization process used 256 samples per epoch, with final evaluation conducted on $256 \times 256$ samples to ensure robust statistical assessment.

### B.3.3 BIOEMU: HOMEODOMAIN

For the homeodomain experiments, we used experimental $^3J$-couplings:

$$^3J_{\mathrm{HN\text{-}HA}}(\phi) = A \cos^2(\phi - \phi_0) + B \cos(\phi - \phi_0) + C, \qquad \phi_0 = 60° \tag{70}$$

using the standard parameterization of Vuister & Bax: $A = 6.98$, $B = -1.38$, $C = 1.72$. Due to high observable covariance, we selected a subset (10/43) of the most informative observables for guidance. This was done via covariance analysis, which identifies redundant measurements that provide overlapping structural information and reveals the effective dimensionality of the conformational space sampled by the observables (see Fig. 10). To identify these observables, we performed PCA on the covariance matrix and identified observables that contribute significantly to multiple principal components (threshold: $|\text{loading}| > 0.25$, see Fig. 11). Determining the Lagrangen mutlipliers was done using Bottaro & Lindorff-Larsen (2018), and the effective sample size was found to be 0.255. BioEmu's internal representation consists of a position matrix $\mathbf{r}$ and a rotation matrix $\mathbf{Q}$ Lewis et al. (2024). Our augmenter module operates directly on the $(\mathbf{r}, \mathbf{Q})$ tuple representation of positions and residue orientations. From these coarse-grained coordinates, we reconstruct the backbone geometry, which allows us to compute experimental observables such as the dihedral angle $\phi$ required for ${}^3 J_{\text{HN–HA}}$-couplings. The augmenter then evaluates the weighted experimental loss and provides its gradients with respect to both $\mathbf{r}$ and $\mathbf{Q}$. For positions, this yields standard Euclidean forces. For orientations, gradients are first mapped to the Lie algebra $\mathfrak{so}(3)$, ensuring that all updates remain consistent with the $SO(3)$ manifold structure. During denoising, the augmenter is evaluated not on the noisy state $(\mathbf{r}_t, \mathbf{Q}_t)$ but on Tweedie posterior mean estimates of the clean structure. For positions, the Tweedie relation

$$\hat{\mathbf{r}}_0 = \frac{\mathbf{r}_t + \sigma_t^2 \, s_t^{(r)}}{\alpha_t} \tag{71}$$

links the network's position score $s_t^{(r)}$ to an estimate of the clean coordinates $\hat{\mathbf{r}}_0$. For orientations, we apply an analogous manifold-aware update in $SO(3)$,

$$\hat{\mathbf{Q}}_0 = \mathbf{Q}_t \exp\!\Big(\widehat{\Omega}\Big(\tfrac{\sigma_t^2}{\alpha_t}\,\omega_t\Big)\Big), \tag{72}$$

where $\omega_t$ denotes the predicted rotational score in axis–angle representation and $\widehat{\Omega}(\cdot)$ maps it to a skew-symmetric matrix. This update guarantees that $\hat{\mathbf{Q}}_0 \in SO(3)$ without requiring an ambient projection. In this way, observable guidance enters the diffusion model consistently for both positions and orientations, while preserving Euclidean and manifold constraints. For training, we used 3,500 samples to evaluate the observable expectations using a batch size of 700. Evaluation was done using 4,900 samples. $\gamma$ was set to 1e-3 and the parameter search space was set to $\eta_{\text{init}} \in [10^{-2}, 1.0]$ and $\kappa \in [3.0, 10.0]$.

### B.3.4 COMPUTE RESOURCES AND RUNTIME DETAILS

Observable guidance experiments were conducted using HPC compute infrastructure equipped with NVIDIA A100 GPUs (80GB memory). Training and evaluation scripts were run on single-GPU nodes.

For the synthetic system (Section 6 and Table 3), each experiment took approximately 4 min to run, consuming 6 GB GPU memory. The chignolin experiments (e.g., Figure 17) required up to 30 min of compute time per run, and 30 GB of GPU memory due to the larger input size and batch requirements. Ablation studies (Figure 14) were conducted with the same hardware and each variant was run across 50 (synthetic case) and 10 (chignolin case) seeds, requiring 1–3 hours per configuration. In total, the reported experiments required approximately 10 GPU hours. Preliminary runs and failed hyperparameter sweeps amounted to an estimated additional 100 GPU hours, not

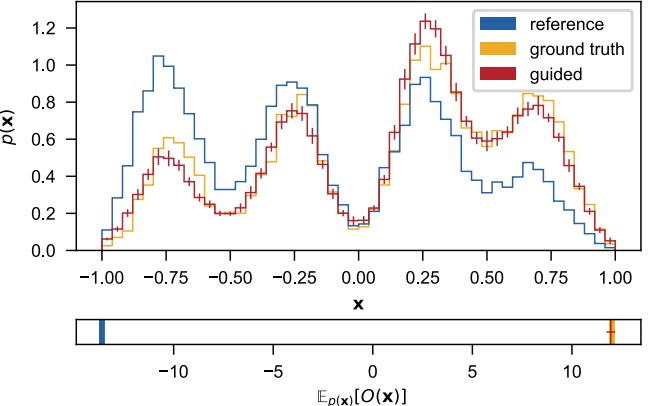

Figure 6: **Comparison of Probability Distributions Before and After Observable Guidance for a 1D Energy Potential.** The top plot shows the probability distributions for three models: the biased reference model (blue), the ground truth model (yellow), and the guided model (red). Guidance helps to align the reference model with that of the ground truth model using only the expectation of an observable function (bottom) while minimizing excess work.

included in the main results. All experiments were executed in a reproducible virtual environment with pinned dependencies (provided in the supplemental code release).

## B.4 PATH GUIDANCE

Similar to observable guidance, path and loss guidance were evaluated on two systems: a synthetic two-dimensional setup and the chignolin mini-protein. We experimented with various functional forms for the guiding strength and time-dependent bandwidth and found that sigmoid-like step functions performed well across both tasks:

$$\eta_t(\vartheta) = \vartheta_{init}\left(1 - \sigma(\vartheta_g(t - \vartheta_s))\right) \tag{73}$$

$$h_t(\varphi) = \varphi_{init} + \sigma(\varphi_g(t - \varphi_s)) \tag{74}$$

To optimize the parameter sets $\vartheta$ and $\varphi$ we applied Bayesian optimization with Gaussian Processes (GPs), using the `scikit-optimize` library. We employed the `gp_hedge` acquisition function, which dynamically combines strategies such as Expected Improvement (EI), Probability of Improvement (PI), and Lower Confidence Bound (LCB) based on their empirical performance. After initial exploration, we restricted the search space to a sensible domain to improve optimization efficiency and support a broader sweep of experimental configurations.

### B.4.1 SYNTHETIC SYSTEM

We evaluated our method on a simple three-moon example, where the two-dimensional dataset consists of three noisy half-moon arcs generated by sampling from shifted semicircles with optional convexity and added Gaussian noise (see Fig. 15). While two of the arcs are well-represented in the training data, only 2.5% of the samples belong to the third arc, creating a challenging low-data region. We adopt the Conditional Flow Matching (CFM) framework (Lipman et al., 2022), from which the score function can be derived for augmentation. To approximate the resulting vector field, we train a four-layer MLP on 10,000 samples for 3,000 steps using a learning rate of $10^{-4}$ and a batch size of 256. Training hyperparameters were selected via a small grid search on an NVIDIA A100 GPU. For optimizing the guidance schedules in Eq. (73), we run 25 Bayesian optimization steps. To classify whether a sample falls within the target moon, we train a two-layer MLP classifier using a learning rate of $10^{-3}$, 1,000 training steps, and a batch size of 256. For path and loss guidance, we evaluated $\gamma$ values between 0 and 1, finding 0.03 working best for path guidance and 0.1 for loss guidance. For sampling we use 20 guiding points generating 1000 samples in one batch.

### B.4.2 CHIGNOLIN SYSTEM

The Boltzmann Emulator used for sampling the chignolin system is described in Appendix B.1. Since loss guidance could not be reliably optimized via Bayesian optimization, we performed an extensive grid search over hyperparameters, including various functional forms for the schedules in Eq. (73). This grid search was run for 24 hours on a single NVIDIA H100 GPU and served primarily to investigate the failure modes of loss guidance. The corresponding results are shown in Fig. 17B. To improve stability, we explored gradient clipping and found it essential for loss guidance. For MEW-guided optimization, we focused exclusively on path guidance. We tested $\gamma$ values between 0 and 1 and found values $\gamma \leq 0.5$ to be effective. Each run consisted of 50 Bayesian optimization steps, with one function evaluation taking approximately 2.5 minutes. As a result, a full optimization run for a fixed $\gamma$ required about two hours on a single NVIDIA H100 GPU (80GB). After each iteration, we computed committor probabilities of the sampled protein conformations using the method described in Appendix B.2 to estimate the proportion of transition-state configurations. For each of the 50 guiding points available, we generated 10 samples, leading to a sample batch size of 500.

## C RESULTS: OBSERVABLE GUIDANCE

All error bars for observable guidance were calculated as the standard deviation between $n$ runs ($n = 50$ for the 1D energy potential experiments and $n = 10$ for the chignolin experiments.

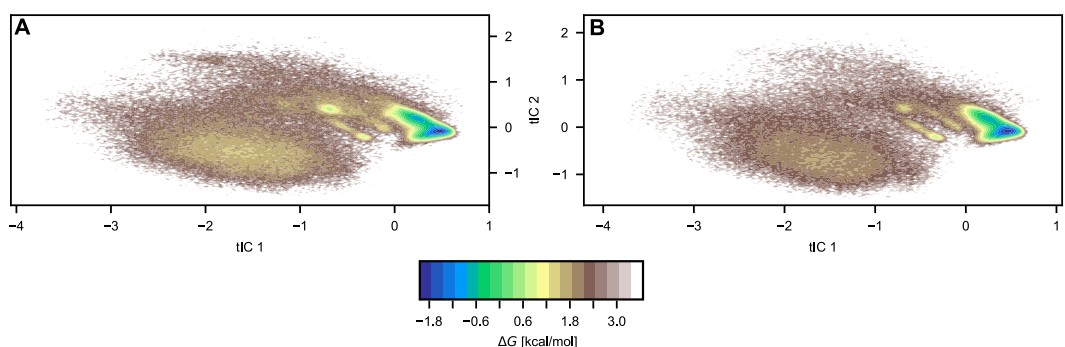

Figure 7: **tICA Projection of Original and Observable-Guided Model.** State space distribution projected onto the first and second tICs for the original (A) and guided (B) BG. The plots are colored by their respective energies.

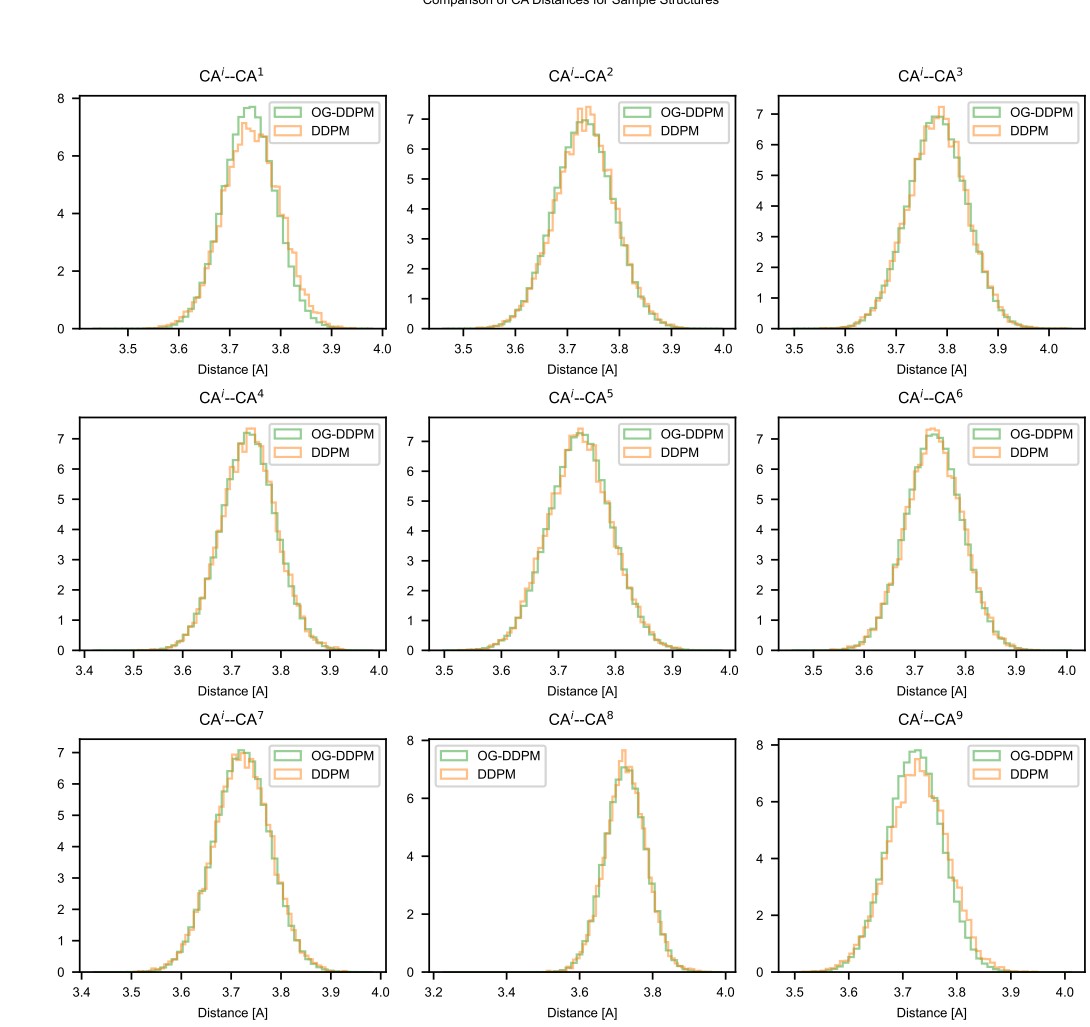

Figure 8: **Comparison of sequential $C^\alpha$–$C^\alpha$ distances between the observable-guided diffusion model (OG-DDPM, green) and the original diffusion model (DDPM, orange).** The plots show the distance distributions for all adjacent $C^\alpha$ pairs (0–2 through 8–9 using zero indexing) in the protein backbone, showing that the guided model maintains proper protein geometry while achieving the desired constraints.

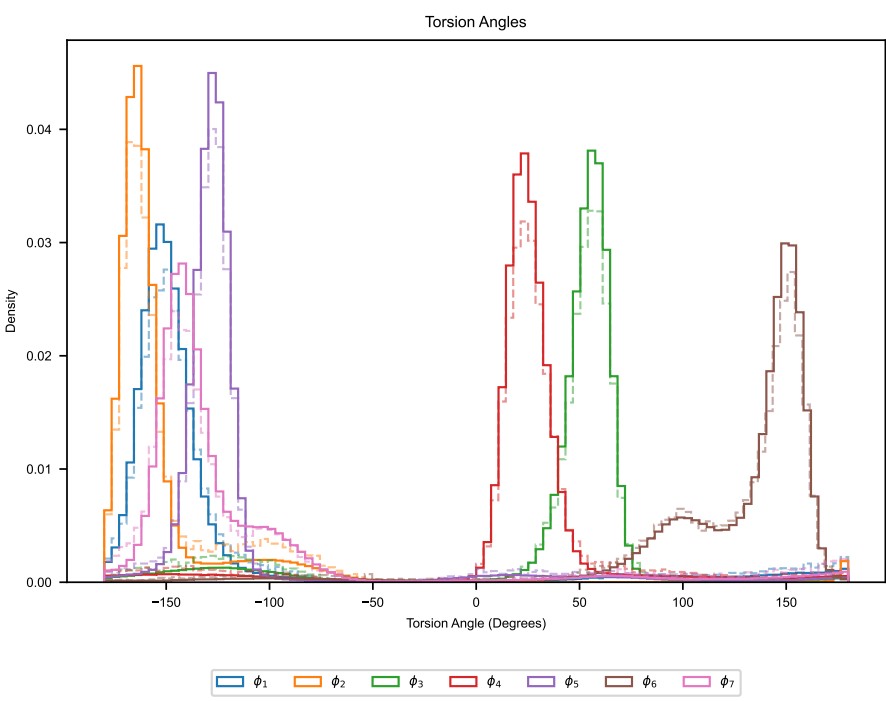

Figure 9: **Distribution of backbone torsion angles ($\phi_1$ through $\phi_7$) comparing the MD simulation (solid lines) with the observable-guided model (dashed lines).** The close agreement between the distributions indicates that the guided model preserves the native conformational preferences of the protein while satisfying the experimental constraints. Each torsion angle is shown in a different color. The differences between the two densities stems from the guidance procedure. Importantly, the the torsion angles themselves remain the same.

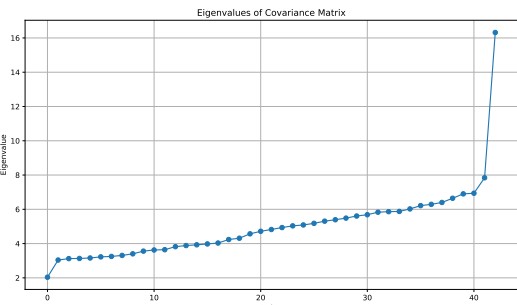

Figure 10: **Eigenspectrum of Observable Covariance Matrix.** The spectrum shows a high correlation between the observables, indicating that most carry redundant information.

## C.1 ABLATION STUDIES

# D RESULTS: PATH GUIDANCE

## D.1 SYNTHETIC SYSTEM

Before applying our method to the Boltzmann Generator on the chignolin system, we first evaluated it on a simple three-moon example (Fig. 15; see Appendix B.4.1 for implementation details). This setup offers a useful testbed, as the low-data region is connected to a high-density area while remaining

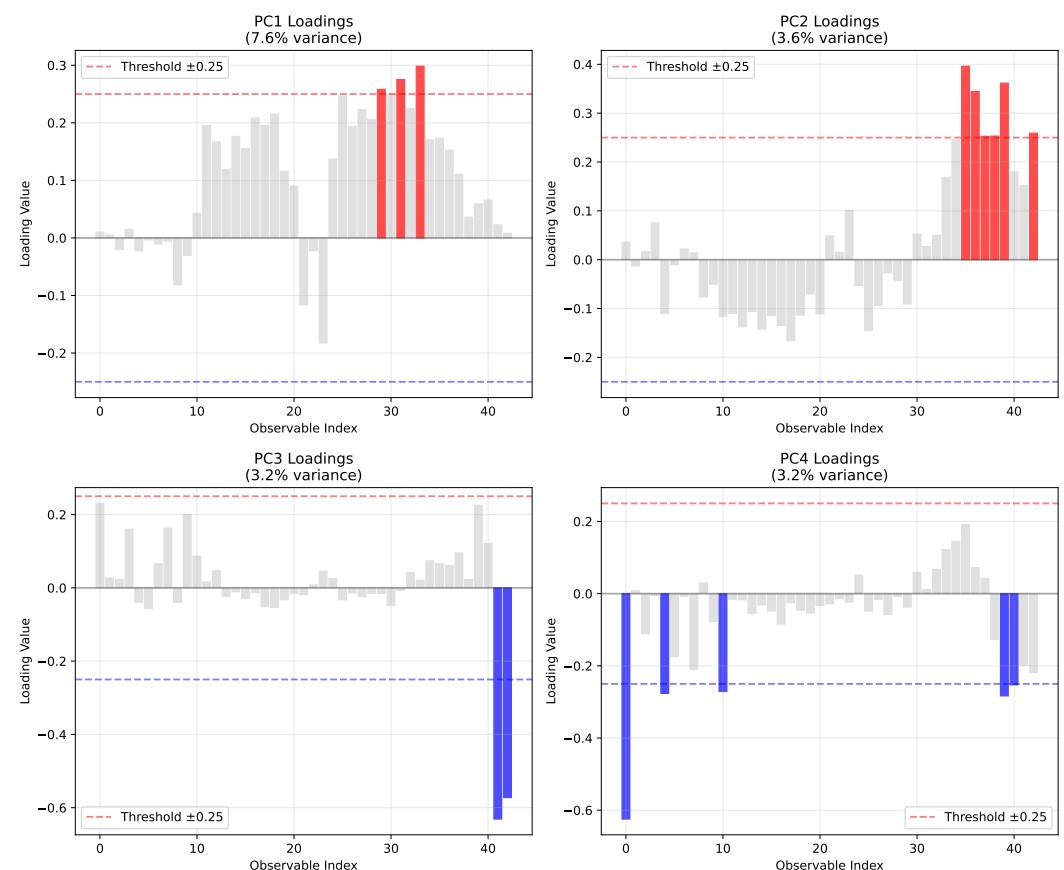

Figure 11: **Principal Components Loading Analysis for Observable Covariance.** The four biggest PCs are shown with a cut-off at 25 % to separate significant contributions to the observables.

Table 3: Metrics for $O(\mathbf{x})$ and KL divergence across models (synthetic).

| Model $\mathcal{M}$ | $\mathbb{E}_{p_{\mathcal{M}}(\mathbf{x})}[O(\mathbf{x})]$ | $\mathrm{KL}(p_{\mathrm{GT}} \| p_{\mathcal{M}})$ |
|---|---|---|
| Ground Truth | 12.01 | — |
| Reference | $-13.6$ | 0.13 |
| Guided | $11.95 \pm 0.22$ | $0.019 \pm 0.002$ |

well-separated from the other half-moon. The objective of guidance in this case is to enable transitions into the low-density region without deviating off the underlying data manifold connecting the moons.

We observe that with ODE sampling, points frequently fall off the manifold, and only careful tuning of the guiding strength minimizes this issue. In contrast, SDE guiding is more robust, as noise helps correct guidance errors. Overall, after minimal optimization of $\eta_t$ and $h_t$, both Path Guidance and Loss Guidance perform well on this toy example. However, in both methods, careful calibration of the guiding strength at low $t$ is essential, as errors at this stage cannot be corrected later. Hence, we found the sigmoid function to be effective in these scenarios, as it naturally converges to 0 for $t \to 1$.

Table 4: Metrics for $O(\mathbf{x})$ and KL divergence with and without MEW regularization.

| Model $\mathcal{M}$ | $\mathbb{E}_{p_{\mathcal{M}}(\mathbf{x})}[O(\mathbf{x})]$ | $\mathrm{KL}[p_{\mathrm{GT}}(\mathbf{x}) \| p_{\mathcal{M}}(\mathbf{x})]$ |
|---|---|---|
| w/o MEW | 0.131 | $0.754 \pm 1.533$ |
| w/ MEW | 0.131 | $0.029 \pm 0.007$ |

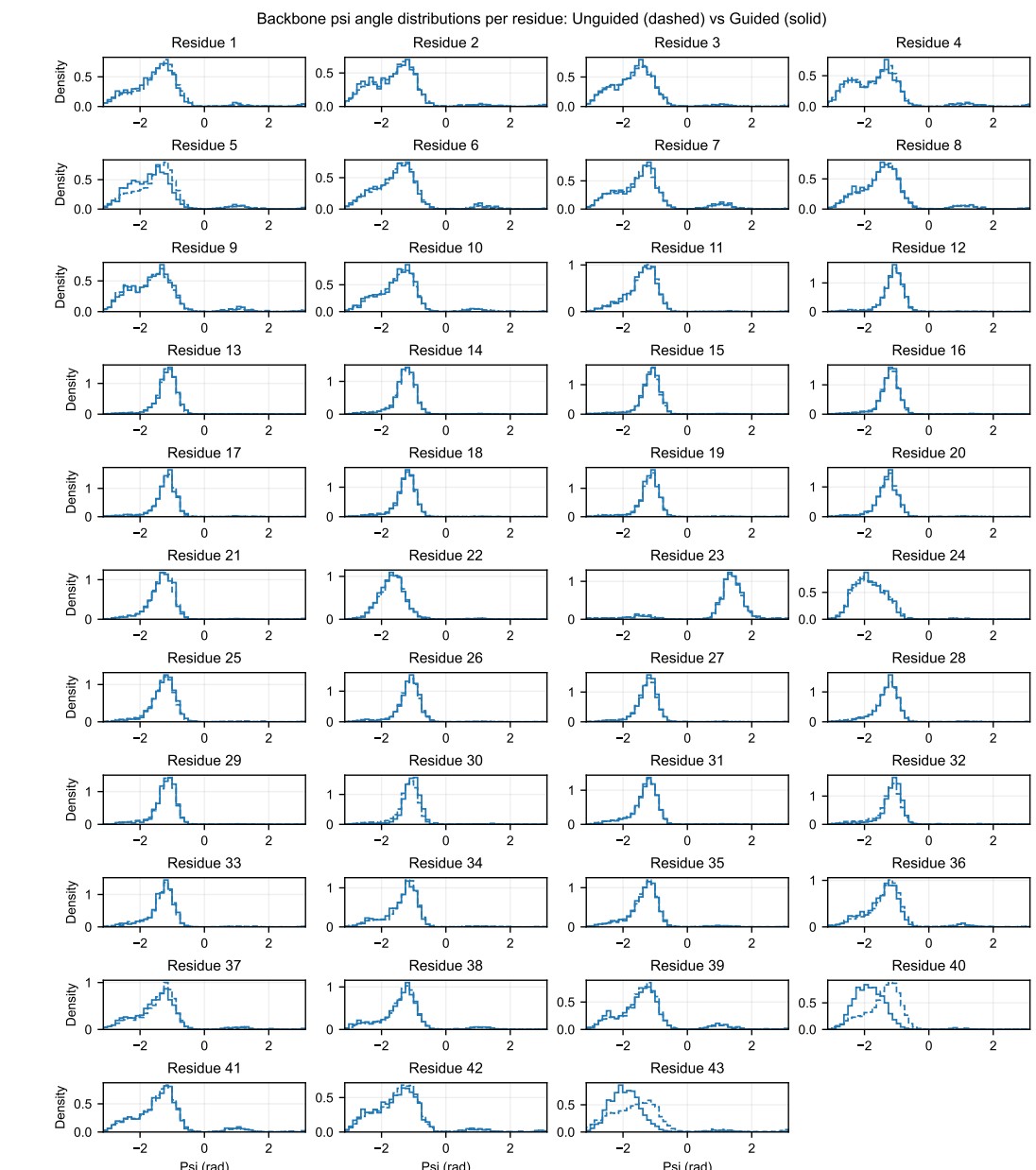

Figure 12: $\phi$ **backbone distribution before and after guidance.** The close agreement between the distributions indicates that the guided model preserves the native conformational preferences of the protein while satisfying the experimental constraints. The differences between the two densities stems from the guidance procedure. Importantly, the the torsion angles themselves remain the same.

In contrast to the Chingolin experiment, we find that loss guidance performs equally well in this synthetic setting, likely due to the simplicity of the data distribution, where the (KDE) in data space sufficiently captures the underlying probability distribution. We also investigate the effect of MEW regularization and observe that omitting the regularization reduces the diversity of the generated samples. Without MEW, the samples tend to be overly guided towards the guiding points on most probable regions, failing to capture the full variance of the underlying distribution Fig. 16.

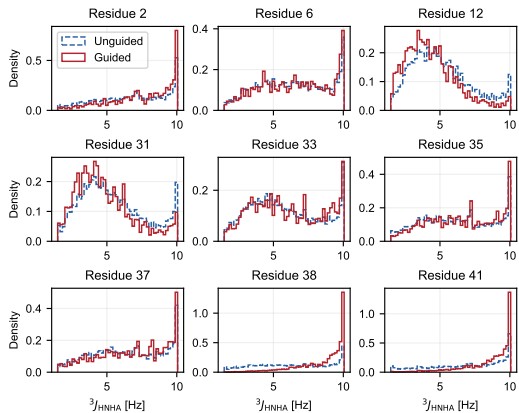

Figure 13: **Distribution of Observables Used in Guidance.** The plots show the distribution of the observables as a function of state space. Blue indicates predictions from the original BioEmu model and red are the MEW-regularized predictions.

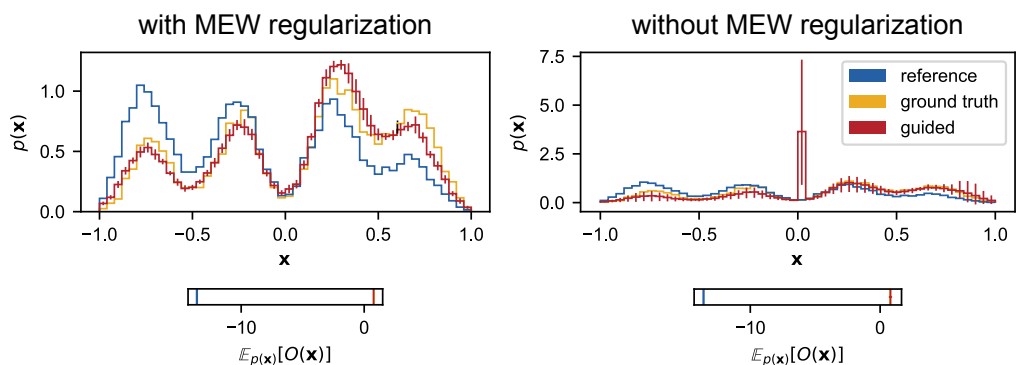

Figure 14: **Ablation study on MEW regularization in the 1D four-well potential.** Left: With MEW regularization, the guided distribution (red) closely matches both the reference (blue) and ground truth (yellow) distributions. Right: without regularization, guidance leads to mode collapse and overconcentration, resulting in low observable prediction error but poor distributional fidelity. Insets show the expected observable values $\mathbb{E}_{p(\mathbf{x})}[O(\mathbf{x})]$.

## D.2 ABLATION STUDIES ON LOSS GUIDANCE

Since reliable sampling with loss guidance could not be achieved, we conducted a more thorough investigation to enable a fair comparison. Instead of relying on Bayesian optimization, we performed an extensive grid search over the guiding parameters (see Appendix B.4.2 for details), with particular focus on smaller guiding strengths to mitigate the effects of unstable or misaligned gradients. Compared to path guidance, the grid search results show substantially lower guiding success, with a maximum transition-state sampling rate of only 0.15%. While this does represent an improvement over unguided sampling (1%), most configurations with non-negligible guidance success resulted in degenerate samples (Fig. 17B). Our analysis suggests that while loss guidance can partially align the model with the target angle distribution, it struggles to follow the desired sampling trajectory throughout the generative process. As a result, strong corrections near the data distribution are required, increasing the risk of sample degeneration (Fig. 17A).

## D.3 BASELINE EXPERIMENTS

In this section, we describe the other two baselines, mentioned in Appendix D.2, which do not augment the vector field. Instead, they utilize the latent representations of the guiding points $\mathcal{X}_t^g$ to initialize the sampling process for generating new points with similar latent characteristics. While

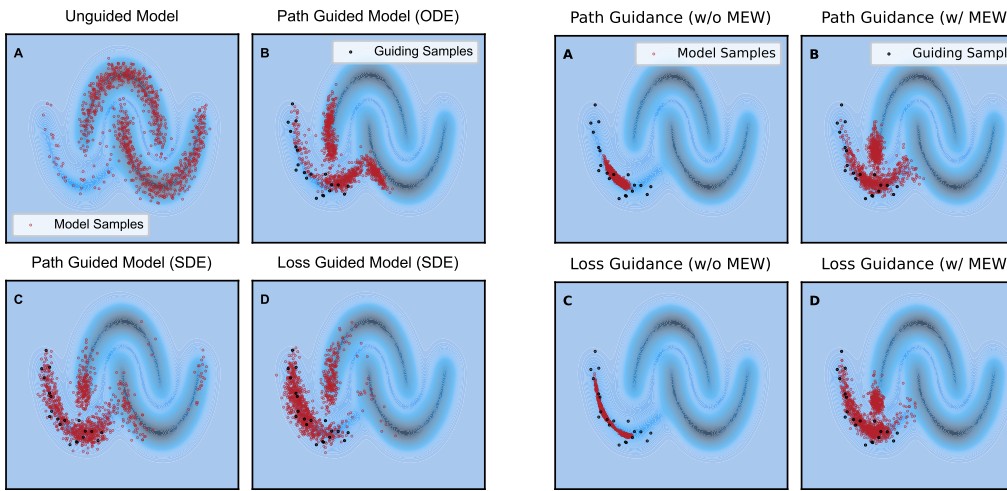

Figure 15: **Sampling the synthetic system.** Comparison of unguided, path-guided, and loss-guided models using both SDE and ODE samplers.

Figure 16: **Guidance with and without MEW regularization**. MEW guidance ensures that we do not collapse onto the guiding points.

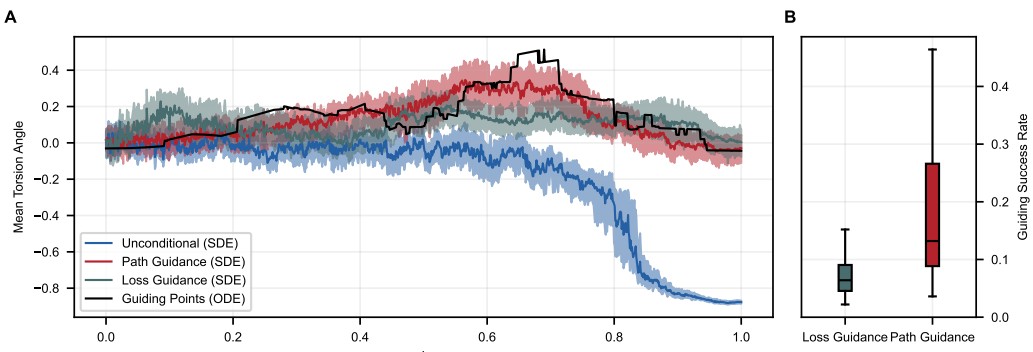

Figure 17: **Path Guidance vs. Loss-Guidance for sampling Transition States**. (A) Evolution of the mean torsion angle (which determines the state of the protein) during the diffusion process. (B) Success rates across different parameter settings.

these methods are appealing in their simplicity, they lack direct control over the sampling process itself.

**Latent-KDE (L-KDE).** We can fit a KDE in the latent space on $\mathcal{X}_1^g$, sample from it, and integrate the probability flow ODE backwards in time. Fitting the KDE at the prior can be advantageous because the Euclidean distance, on which most kernels are based, is better suited for Gaussian-distributed data compared to its use in data space. We refer to this method as Latent-KDE (L-KDE).

**Stochastic-Reverse (SR).** Alternatively, we can select a specific time step $t$ such that the desired properties are preserved and initialize the backward SDE (Eq. (1)) with latents from $\mathcal{X}_t^g$. The stochasticity of the SDE will ensure we generate new divers samples with $\mathbf{x}' \in A$.

We conduct sampling experiments using the aforementioned baseline methods to verify whether the results align with our intuition. Specifically, for the L-KDE baseline, we evaluate a Gaussian kernel with noise levels (standard deviations) of $\{0.01, 0.05, 0.1\}$. For the SR baseline, we consider intermediate times $\{0.1, 0.5, 0.9\}$. For simplicity, we only examine the scenario where there is a single guiding point (i.e., $\mathcal{X}_1^g$ and $\mathcal{X}_t^g$ are singleton sets). Each experiment is repeated with five different seeds.

In the following figures (Figures Fig. 18 – Fig. 23), we provide various metrics, histograms, and energy surfaces that summarize the trends observed in these baseline guidance scenarios. Overall, the results strongly suggest that guidance biases the sampling procedure toward the reference guiding points, which aligns with our intuition.

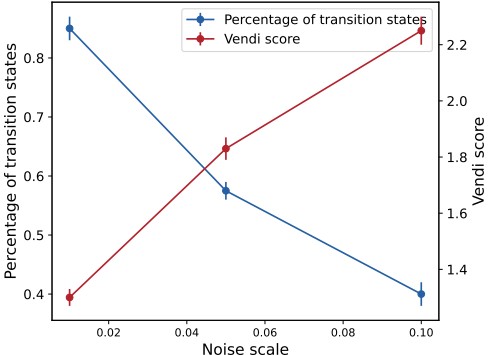 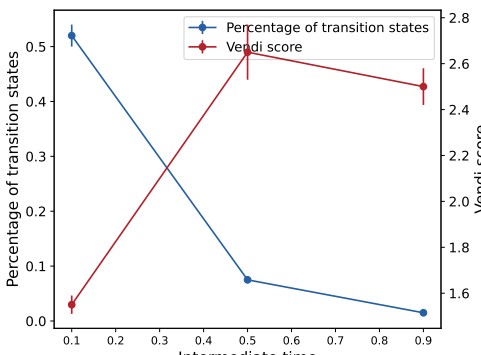

Figure 18: **Trade-off between sample variance and guidance success rate (L-KDE).** As the KDE noise scale increases for the L-KDE baseline, the percentage of transition states among the generated samples decreases (blue), while the vendi score among the generated states increases (red).

Figure 19: **Trade-off between sample variance and guidance success rate (SR).** As the selected time step $t$ increases for the SR baseline, the percentage of transition states among the generated samples decreases (blue), while the vendi score among the generated states tends to increase (red).

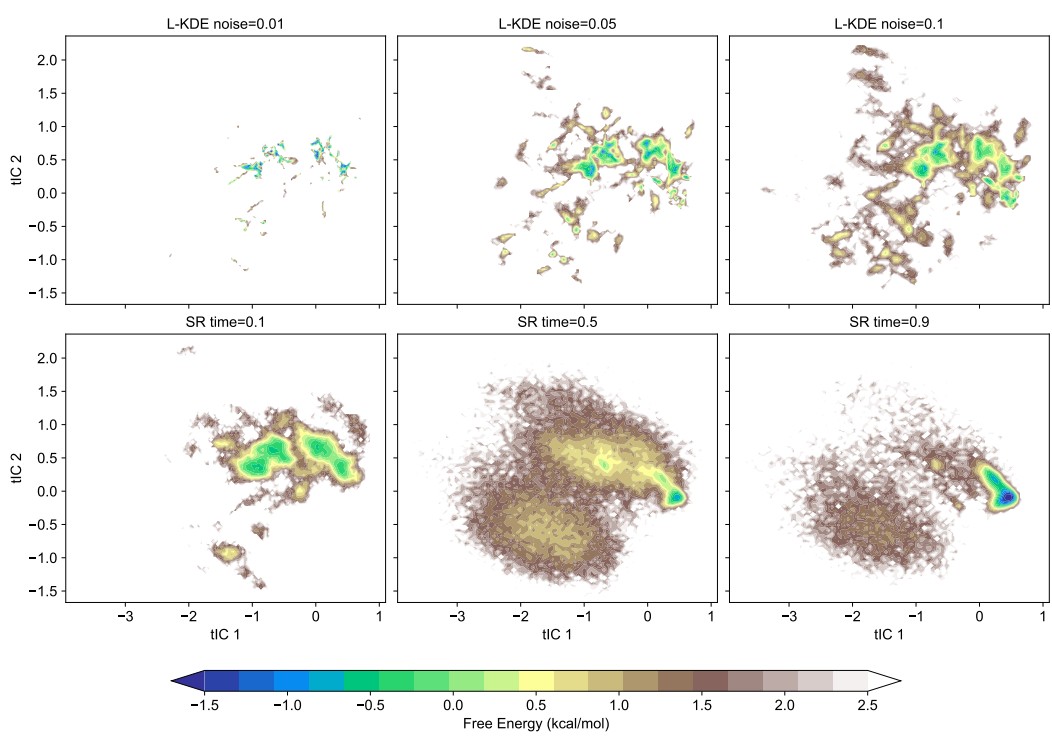

Figure 20: **Energy surface plots.** First row: L-KDE baseline for various levels of noise scale. The smaller the perturbation, the more concentrated the samples around the transition states region. Second row: SR baseline for various values of intermediate time. The smaller the stochasticity level, the more concentrated the samples around the transition states region. Compare with Fig. 5 and Fig. 2 A and B.

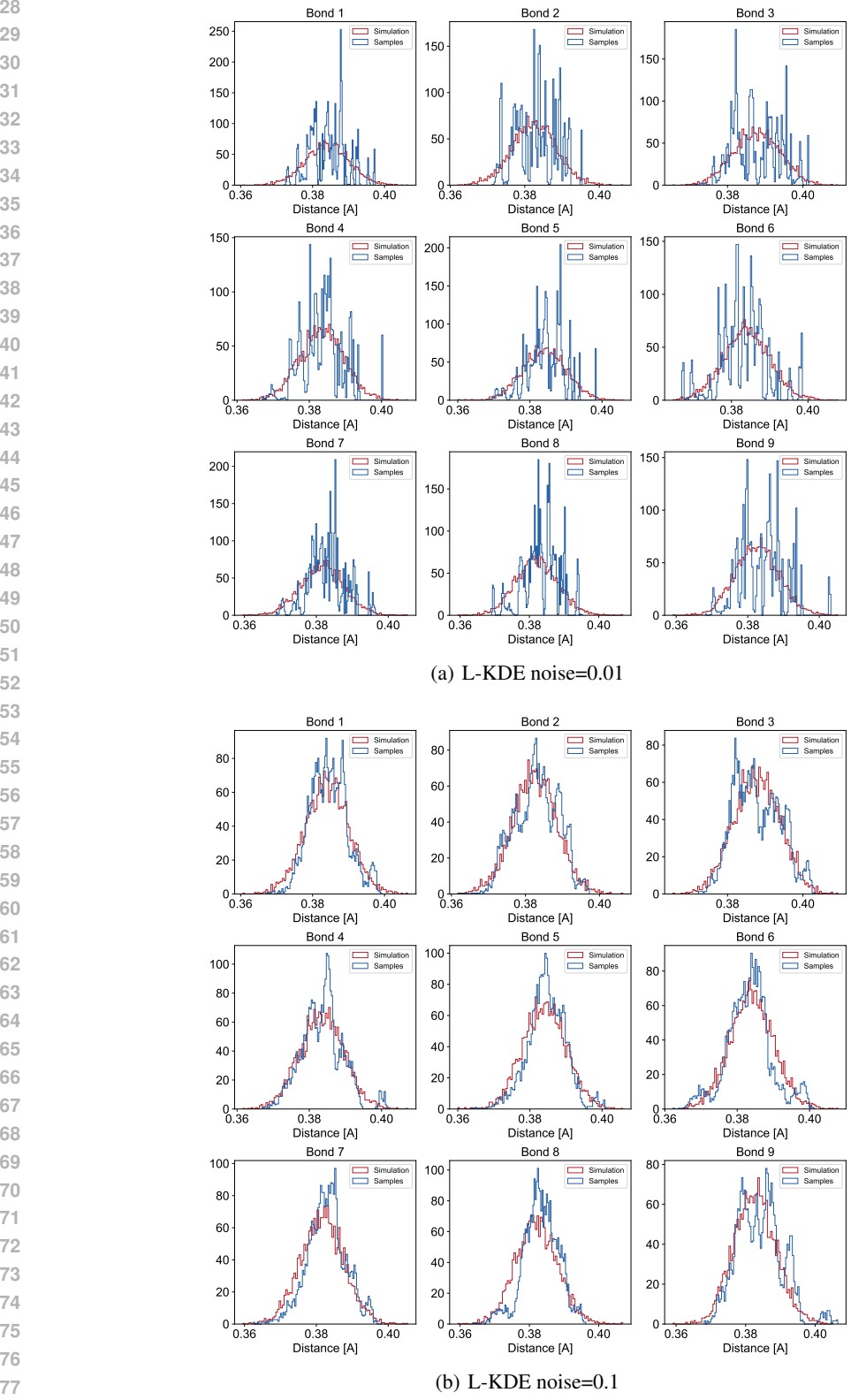

Figure 21: **Comparison of bond distance distributions for L-KDE and the reference.** The L-KDE baseline (blue) is superposed on the corresponding histogram of the unconditional (red) distribution (the CLN025 MD simulation). We see that for small perturbations, the generated samples seem to conform to particular details of the guiding samples. As the noise increases, the guidance impact diminishes. This is quantified in a more principled way in Fig. 23.

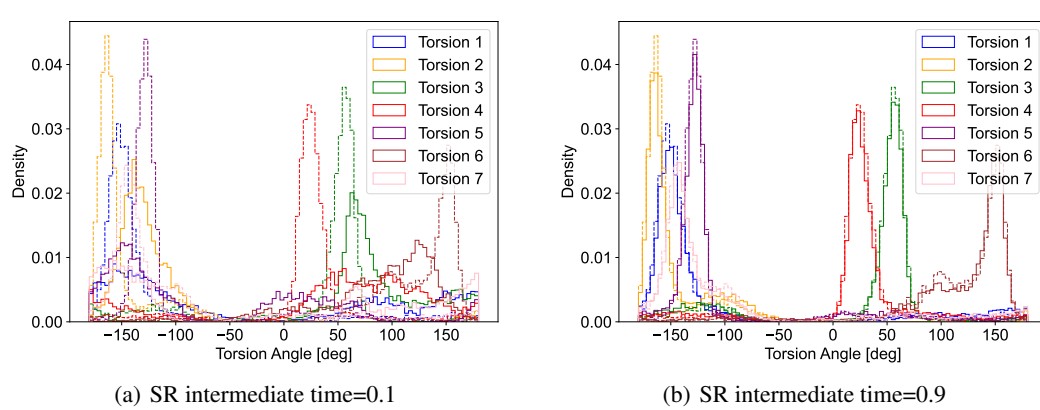

(a) SR intermediate time=0.1        (b) SR intermediate time=0.9

Figure 22: **Torsion angle histograms for the SR baseline at different noise levels.** Solid lines show SR samples at $t = 0.1$ (Left) and $t = 0.9$ (Right), superimposed on the corresponding unconditional distributions (dashed lines). At high stochasticity ($t = 0.9$), the torsion angle distribution becomes nearly indistinguishable from the unconditional one.

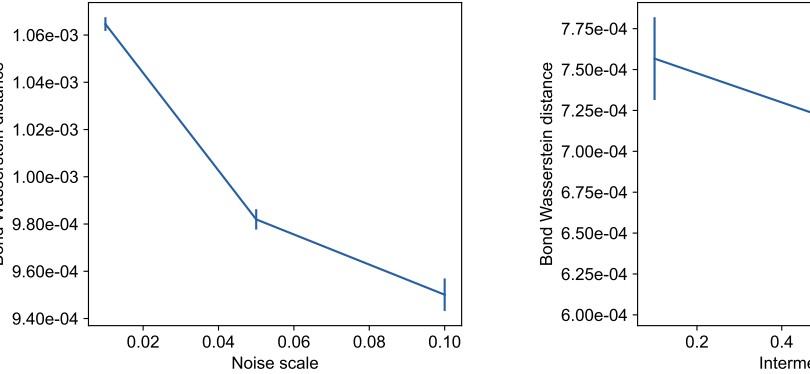

(a) Wasserstein distance vs noise scale for L-KDE baseline

(b) Wasserstein distance vs intermediate time for SR baseline

Figure 23: **Wasserstein distance between baselines and reference bond distance distributions.** We measure the distance between the bond distance distributions of baseline methods and the CLN025 MD simulation (see Fig. 21). As the stochasticity level increases for both baselines, the generated distributions converge toward the unconditional reference, indicating a reduced influence of the guidance signal.

