# OpenReview forum: "Minimum-Excess-Work Guidance"
_ICLR.cc/2026/Conference — ICLR 2026 Conference Withdrawn Submission_

### Official Review · Reviewer_HfJW · 2025-10-27

**Soundness:** 3
**Presentation:** 4
**Contribution:** 2
**Rating:** 4
**Confidence:** 4

**Summary:**

The authors propose a framework to guide probability flow generative models using a minimum excess work (MEW) framework. They motivate their method physically and theoretically and evaluate it on synthetic and protein data. They show that MEW is able to improve estimation of observables and sample rare event regions.

**Strengths:**

- The paper is clearly written and well-motivated
- The theoretical justification motivates the proposed guidance and regularization technique
- The method clearly improves observable estimation

**Weaknesses:**

- The experiments lack simple baselines. Including some sort of simple classifier guidance and showing that is fails due to lack of regularization would strengthen the paper.
- Many of the experiments seem to report improvements in observable estimation for the same observables used for guidance (e.g. 4.1.2). Are the JHN-HA couplings included in the 10 experimental observables used for guidance in 4.1.3? A clearer discussion surrounding which observables are used for guidance, and to what extent guidance improves estimation of other observables, would make the paper stronger.
- Where do the samples for transition state guidance come from? If the goal is to sample rare events, then clearer discussion surrounding to what extent some knowledge of the transition region is required would make the paper stronger. It would also be interesting to compare to other existing literature to ground the benefit of the proposed method.

I am happy to raise my score if some of my points are addressed.

**Questions:**

Please see above.

---

### Official Review · Reviewer_Cwj8 · 2025-10-31

**Soundness:** 2
**Presentation:** 3
**Contribution:** 3
**Rating:** 8
**Confidence:** 3

**Summary:**

The method aims to perturb learned (diffusion) generative models by introducing a new score term into their diffusion process. In order to avoid straying too far from the "data manifold," a regularization is introduced whereby the excess work (work done more than is necessary to generate the learned distribution) is penalized. The justification comes from the fact that kullback-leibler and wasserstein divergences have excess work appear explicitly in their formulations. The methods proposed focus on changing the distribution such that observables match experiment or some (transition) regions are better sampled. It's written well, simple to follow, although the justification is not very rigorous it agrees in spirit with many recent works on the subject.

The results include observable and path guidance on synthetic data, observable gudiance with coarse grained chignolin and bioemu, finally path guidance on coarse grained bioemu energy for chignolin.

**Strengths:**

# general
- the method addresses interesting problems for the current moment (reweighing for experimental observables and finding transition states)
- clearly presented with strong empirical results, albeit with minimal comparisons.

**Weaknesses:**

# method
- the need for a kernel to identify transition regions seems extremely limiting in practice, i.e. one would need a good representation of the transition state and a similarity kernel which probably does not scale well with dimensionality.

# general
- the paper lacks very strong comparisons to baselines. There are other methods to try; however, each of them have not been applied to the specific cases covered in this work. That increases the cost of comparison; however, it would be in the spirit of a NeurIPS paper.

**Questions:**

- Is path guidance extremely expensive? It seems like it would be.
- It seems that evaluation of path guidance is generally weaker. You claim that multimodal transition states would pollute the signal and weaken the guidance. Is it possible to justify this claim further?
- I think comparing with other methods is most appropriate for path guidance where your baseline "loss guidance" has many caveats (as you fairly mention). Can you comment on the difference between your method and, e.g., adjoint matching for this task. It seems like something like that would be a fairer comparison.

---

### Official Review · Reviewer_fSQ3 · 2025-11-03

**Soundness:** 2
**Presentation:** 3
**Contribution:** 2
**Rating:** 2
**Confidence:** 3

**Summary:**

The paper proposes a framework for algorithms steering flow-based generative models (e.g. DDPM, normalizing flows) through the minimal work principle often used in control theory and statistical mechanics. The motivation here is to keep the new distribution as close as possible to the original distribution. Experiments are presented on low-dimensional synthetic data, coarse-grained chignolin, and BioEmu.

**Strengths:**

- The objective is clear, easy to understand, and straightforward to implement. Simplicity also means there are minimal hyperparameters to tune.
- Theory is also clear and fleshed out well, helping future practitioners understand the role of this approach.


- Many ablations and various metrics on the method also help future practitioners understand the method better.

**Weaknesses:**

- There is a notable lack of baselines. The objective is very similar to other stochastic control–style fine-tuning methods that employ this minimal work idea from optimal control and stat mech, e.g. [Adjoint Matching](https://arxiv.org/abs/2409.08861) (see e.g. eq. (28)), as well as the already extensive body of work on guiding diffusion models and flow matching models towards a new energy function (e.g. anything learning the tilted distribution $p_{\text{data}} e^r(x)$).


- Success depends on many different factors (KDE design, time-varying bandwidths, optimization schedule) that are not ablated or discussed for sensitivity.

- The specific kind of divergence used in this paper (L2) is not compared, theoretically or empirically, to alternative kinds of divergence used in related work (e.g. KL). See eg. [Tang and Zhou](https://arxiv.org/pdf/2403.06279).

**Questions:**

- How does the method perform against Adjoint Matching and other SOC-style approaches? Are there particular benefits to the proposed approach?


- Are there any settings presented in the paper where the main diffusion guidance baselines (e.g. Diffusion-DPO, adjoint matching) are inapplicable?


- Are there unique theoretical advantages from using the L^2-based divergence over an alternative like KL (e.g. does stability not hold for KL)?

---

### Official Review · Reviewer_pkaq · 2025-11-11

**Soundness:** 3
**Presentation:** 3
**Contribution:** 2
**Rating:** 4
**Confidence:** 4

**Summary:**

The authors look at formulating an “excess work” term (based on the principles of minimum excess work) as a regularizer to guide generative models. This involves adding an additional perturbation to the score model. The generative models are guided based on either observables (to align the generated distributions), or by aligning the path of the generative model. The authors look at a synthetic data example and coarse-grained proteins to test this idea.

**Strengths:**

- The paper is mostly easy to read and follow.

- The general motivation of such a framework, using experimental observables, makes sense, and is one worth looking at by the community.

**Weaknesses:**

- It seems like the guidance is being done with an observable, and then the ML model prediction is also tested on that observable (such as in section 4.1.2). This doesn’t seem like a fair comparison setting, as you are predicting something that you’re guiding on. How do you do in cases where the observable is held-out? The evaluations of the method on the two protein cases could be more comprehensive.

- The authors look at path guidance to compute transition states. This section seems ad hoc in nature in terms of what and how the authors are trying to apply MEW regularization, and Figure 4 isn’t a comprehensive metric for quantifying sampling these transition states. More evaluations would be helpful here, such as computing transition rates or free energies. Additionally, this is a problem that has been studied in the past using ML, and the authors do not make any comparisons to other baseline ML methods for this task (they use loss guidance as their baseline).

**Questions:**

- How is the reference model in Section 4.1.2 defined?

- In section 4.1.2, are you both guiding with an experimental observable (and making predictions on that observable), and also generating new structures of chignolin?

- Can you provide more details on the exact amount of data samples that are used to train the model? I found this hard to find (except in section 4.1.3, where it seems like 10 experimental data points were used for guidance).

In section 4.1.3, is the unguided model a pre-trained BioEmu model?

- in section B.3.3, the authors mention that the effective sample size for this example is 0.255. Could the authors provide more clarification on what this is referring to? This seems low, and could make estimates unstable.

- Is there some calibration that needs to be applied to account for any uncertainty or noise in the observables?

- Given that the authors are using a supervised observable loss, and assuming a sparse data setting, how do you ensure overfitting is prevented?

- There are some works that seem related to the general idea presented in this paper, where they have a formulation to find the minimum energy path to sample between states. How does this work compare to these approaches?
> [1] Peterson and Covino. PINN-MEP: Continuous Neural Representations for Minimum Energy Path Discovery in Molecular Systems. FPI-ICLR (2025). [2] Raja et al. Action-Minimization meets Generative Modeling. Efficient Transition Path Sampling with the Onsager-Machlup Functional. ICML (2025)

---

### Note · Authors · 2025-11-18

I have read and agree with the venue's withdrawal policy on behalf of myself and my co-authors.